# Learning with Coupled Uncertainty

## Abstract

We initiate the study of decision-making under coupled uncertainties. In this problem, a learner has access to ground truth and coarse measurements of outcomes and would like to use them for decision-making. The learner has constrained access to ground truth measurements for only a given fraction of decision outcomes and would like to leverage the cheaper coarse measurements of decision outcomes. We introduce a model where the randomness of the ground and coarse measurements is coupled, and our approach learns their correlation to optimally combine coarse measurements with ground truth and achieve improved performance. This framework unifies several settings, like learning from multi-fidelity data sources and delegating decision-making to AI agents. We provide an upper confidence bounds based algorithm CUUCB for leveraging coupled uncertainties in a multi-armed bandit task, where the covariance structure between coarse measurements and ground truth is unknown. We show theoretically how CUUCB adapts to the underlying covariance structure by deriving instance-dependent and instance-independent regret bounds. We validate our algorithm in two experiments: a task with synthetically generated data, and an LLM benchmarking task. We compare our algorithm to existing UCB variants with access to only ground truth measurements on the constrained fraction of outcomes. In both cases, our algorithm is able to achieve lower regret.

## 1 Introduction

Recent advances in artificial intelligence (AI), computational modeling, and the development of foundation models for science are reshaping the way experiments are conducted — accelerating the pace of new discoveries Bommasani et al. (2021); Wang et al. (2023). Predictive tools and language models are increasingly being developed in domains like personalized healthcare Lu et al. (2021); Yang et al. (2022); Thirunavukarasu et al. (2023), polling Tumasjan et al. (2010); Wang et al. (2015), environmental monitoring Shi et al. (2017); Ravuri et al. (2021), and content moderation Gomez et al. (2024) paving the way for new services and more efficiencies in important application areas. Despite these advances, these powerful predictive models are often *black-box* which introduces challenges when one would like to use them for downstream decision-making tasks. For example, language models may be mis-aligned with the underlying human preferences one would like them to emulate. Despite a large body of recent work on uncertainty quantification for black-box machine learning predictions (the exemplar being conformal prediction Vovk et al. (2005); Romano et al. (2019); Angelopoulos et al. (2023b)), existing methods often yield overly conservative estimates or are simply not amenable to online decision-making.

To overcome this challenge, we propose an algorithm for sequential decision-making when one has access to two different types of measurements: (1) coarse measurements (e.g., machine learning or AI-based predictions) which may be cheap and readily available, and (2) ground-truth measurements that may require costly experimentation. The decision-maker does not know how good the coarse measurement is but they would like to minimize their regret over a sequence of decisions while keeping the number of "ground-truth" measurements they query below a desired threshold. To tackle this, we introduce a feedback model where (1) the randomness in the coarse and ground truth measurements is *coupled*, and (2) each time a ground-truth measurement is queried, the corresponding coarse measurement is also observed at negligible cost. This framework captures feedback mechanisms that naturally arise in a variety of empirical settings, including fluid dynamics modeling, climate forecasting and active learning epidemiology systems Niu et al. (2024); Wu et al. (2023; 2022); Li et al. (2022a; 2020).

The key premise of *coupled uncertainty learning* is that if the coarse measurements are highly correlated with the ground-truth measurements, then one can use them to reduce the variance of the estimators obtained from only ground-truth measurements and ultimately to substitute for ground truth measurements altogether. Doing so however, is not straightforward since the covariance structure of the ground truth and coarse measurements may not be known a priori, and consequently, the weight that should be given to the coarse measurements in estimation is unknown. Furthermore, high correlation between ground truth and coarse data is not enough to simply replace one dataset for another since it does not solve the problem of *mis-alignment*—a problem that our experiments on large language models clearly expose.

In this paper, we initiate a study of *learning* with coupled uncertainties in the context of multi-armed bandits. We seek to learn the optimal arm while building estimators that combine ground truth and coarse data to achieve lower regret *with respect to a ground-truth baseline* than existing algorithms. Reflecting the fact that in many applications ground truth feedback can be expensive to collect (e.g., querying human feedback in language-model benchmarking tasks can be costly), we constrain the number of ground truth queries to be an $\alpha(i)$ fraction of the total number of pulls for a given arm $i$. This ratio can arise from the relative costs of ground truth vs coarse experimentation.

We propose the Coupled Uncertainty Upper Confidence Bounds (CUUCB) algorithm, which can be seen as a variant of the UCB-V algorithm (Audibert et al., 2009), which is a variance adaptive upper confidence bounds algorithm. We show that CUUCB achieves an instance-independent expected regret bound of $\mathcal{O}(\sqrt{C_\alpha T \log T})$ while only requiring approximately $\alpha(i)$ fraction of pulls for arm $i$ being ground-truth samples. Here, $\alpha(i)$ is a user-defined parameter and can be interpreted as the target fraction of pulls of arm $i$ which access ground truth. $C_\alpha$ depends on the covariance structure of the ground-truth and coarse measurements $\alpha$ and has the following structure:

$$C_\alpha = \sum_i \mathrm{Var}^{\mathrm{G}}(i) \left( \frac{(1 - \rho(i)^2)}{\alpha(i)} + \rho(i)^2 \right) \tag{1}$$

where $\mathrm{Var}^{\mathrm{G}}(i)$ is the variance of the ground-truth measurements for arm $i$ and $\rho(i) \in [-1, 1]$ is the correlation between ground-truth and coarse measurements for arm $i$. This can provide improved performance compared to using a variance adaptive algorithm such as UCB-V (Audibert et al., 2009), when there is no access to coarse measurements and the regret bound has the form $\mathcal{O}(\sqrt{C'_\alpha T \log T})$ with $C'_\alpha = \sum_i \frac{\mathrm{Var}^{\mathrm{G}}(i)}{\alpha(i)}$. We can thus expect to see improved performance when $|\rho(i)|$ is large, meaning the ground-truth and coarse measurements have a high correlation, and when $\alpha(i)$ is small, meaning only a small fraction of the measurements are ground-truth.

We verify our findings experimentally both on synthetic data and a LLM benchmarking task in which we observe that our algorithm can overcome mis-alignment in the LLM-as-a-judge paradigm using small amounts of ground-truth data.

## 2 RELATED WORK

Extensive research since Robbins (1952) has studied the multi-armed bandit problem, with particular emphasis on algorithms using upper confidence bounds (Auer, 2002; Bubeck et al., 2012; Lattimore & Szepesvári, 2020). Within this literature, the most relevant to this paper is a line of work on multi-fidelity bandits (Kandasamy et al., 2016b;a) in which the authors consider bandit problems when multi-fidelity information is available. In such problems, only only one type of feedback (or fidelity level) is available at each time step, and the fidelity levels are known. In contrast our coarse measurements are of unknown quality. By introducing the framework of coupled uncertainty bandits, we highlight new way in which multiple fidelity feedback can be leveraged for decision-making. By collecting *both* low and high fidelity feedback when querying the higher fidelity levels, the measurements become statistically coupled. This allows the learner to learn correlations and improve performance over and beyond existing bandit algorithms that use only high fidelity data.

These problems have also surfaced in the literature on adaptive experiment design for scientific applications of machine learning (Fernández-Godino et al., 2016; Greenhill et al., 2020) where the goal is to fuse data sources with multiple fidelity levels and costs—online— to derive good estimates of desired quantities. Previous works have focused on Bayesian optimization and Bayesian active learning problems, aiming to iteratively optimize or learn a target function by leveraging data across

different fidelity levels (Li et al., 2020; Hernandez-Garcia et al., 2023; Li et al., 2022a; Kandasamy et al., 2017). One popular approach involves using Gaussian processes (GPs) for multi-fidelity surrogate modeling, providing well-calibrated uncertainty estimates for acquisition function design (Wu et al., 2020). Other multi-fidelity surrogate modeling approaches utilize neural processes (Wu et al., 2022; 2023) or ordinary differential equations (Li et al., 2022b) as an more flexible and scalable alternative to GPs. None of these lines of work consider the potential effects of leveraging the coupling between data sources to reduce uncertainty.

Our work also relates to the emerging literature on using AI predictions, large language models, or other machine learning based models as substitutes for experimentation in estimation problems. In offline settings, these problems have recently emerged under the umbrella of *prediction-powered inference* (Angelopoulos et al., 2023a; Zrnic & Candès, 2024) which provides a generalization of classic estimators studied in statistics like the augmented inverse propensity weighted (AIPW) estimator from (Robins & Rotnitzky, 1995). These works leverage predictive models to improve the efficiency and accuracy of inference, particularly in settings where labeled data is scarce or expensive to obtain. These techniques have been used to deal with problems of mis-alignment and a lack of human data in the fine tuning of large language models (LLMs) (Zhou et al., 2025). Our paper extends some of these ideas to problems of online learning. Concurrent work (Ji et al., 2025) has also extended these ideas to sequential decision-making settings in an attempt to warm start online learning algorithms with coarse data. Our paper looks at a different problem of extracting statistical signal from coarse data while collecting data under a budget constraint. Furthermore, our theoretical results and derivations gives finite-sample bounds on estimators under a general boundedness assumption that makes the derivations nontrivial. These expand upon the more asymptotic analyses present in the literature on prediction-powered inference and may be of independent interest.

## 3  THE COUPLED UNCERTAINTY FRAMEWORK

We start with formulating the paradigm of decision-making with coupled uncertainty. In this setting, we assume that there is a sequence of contexts $\{c_t\}_{t=1}^{\infty}$ arriving into the system. Given a context, $c_t$ at time $t$, the system chooses an action $i_t$ from a set of actions $\mathcal{A}$, generating an outcome $y_t(c_t, i_t)$. These contexts and actions might be (patient, treatment) pairs in medical scenarios or (prompt,LLM) pairs in language model triage tasks. Importantly, we assume that outcomes can be measured with different levels of granularity. For simplicity of exposition, we assume that there are two such levels of measurement, one coarse and one ground-truth against which we would like to be benchmarked.

At time $t$, we formally denote $r_t^C(y)$ as the coarse measurement we receive given the outcome $y$, and $r_t^G(y)$ as the ground truth one. To further concretize this setup we present a motivating example based on benchmarking large language model that we experimentally validate our algorithm on in Section 6. We remark that many other use cases including personalized medicine, scientific discovery, content moderation, and polling fit into this framework.

**Example 3.1** (LLM-as-a-judge). *A number of existing platforms benchmark and rank the performance of large language models (LLMs). This is then used to either keep leaderboards of language model performance (Chiang et al., 2024), or ultimately to triage prompts to optimal models online (Ong et al., 2025), i.e., to perform model routing. The state-of-the-art approach is to allow LLM users to compare and rate the LLM outputs, and match decisions to those made by humans. However, high-quality human-level data is scarce, relative to the categories of the prompts we are interested in and asking people to provide data can be costly. Therefore many researchers make use of language models to facilitate and provide AI feedback. This raises the question of whether the AI feedback is comparable or* aligned *to human preferences and also when and how to make use of it. Such problems fit neatly into our framework: one would like to make use of LLMs as judges of quality (i.e., coarse feedback) and use them to replace human-feedback (i.e., the ground truth). At each round $t$, given the prompt $c_t$ and the LLM choice $i_t$, the benchmarking algorithm can either query just the coarse level AI feedback $r_t^C(y_t)$ and save human effort, or query both the ground truth human feedback and the AI feedback as a tuple $[r_t^G(y_t), r_t^C(y_t)]$. Existing LLM-as-a-judge Gu et al. (2025) approaches can easily fall prey to mis-aligned models. As we observe experimentally, incorporating even small amounts of ground truth data ( 5%) allows us to overcome this problem.*

In many examples, we observe that when the decision makers have access to fine-grained ground truth data $r_t^G(y)$, the coarse level measurement $r_t^C(y)$ with the same input $y$ is cheap to obtain. We therefore

assume in our protocol that when the learner opts to observe a ground truth measurement, they receive a two-dimensional tuple $[r_t^{\mathrm{G}}(y), r_t^{\mathrm{C}}(y)]$. Otherwise, they receive a single coarse reward value $r_t^{\mathrm{C}}(y)$. Our framework assumes that the context $c$ arises from some underlying distribution and treats it as a random variable that causes coupling of the ground truth and coarse measurements. Thus, the distribution of $[r_t^{\mathrm{G}}(i, c), r_t^{\mathrm{C}}(i, c)]$ depends only on the learner's action $i$ as $c$ is a random variable. We thus omit the context $c$ in the following discussion since it is understood to be part of the randomness. We remark that a natural extension of our framing would be to consider context-dependent algorithms, though we leave such problems to future work.

In the remainder of the paper we focus on multi-armed bandit problems and demonstrate in our analyses that we can utilize the coupling between the measurements to obtain tighter concentration bounds for the means which ultimately resulting in performance gains.

## 4 MULTI-ARMED BANDITS WITH COUPLED UNCERTAINTIES

In our analysis, we study the multi-armed bandit problem where at time $t$ the learner plays an action $i_t \in \mathcal{A} = [K]$. The learner also has a choice of what type of reward measurement $m_t$ to observe at time $t$, a coarse and a ground truth measurement ($m_t \in \{\mathrm{C}, \mathrm{G}\}$). We assume the existence of joint measurement vectors $r_t^{\mathrm{C}}, r_t^{\mathrm{G}} \in \mathbb{R}^K$ such that the conditional mean reward values satisfy

$$\mathbb{E}[r_t^{\mathrm{C}}(i)] = \mu^{\mathrm{C}}(i), \qquad \mathbb{E}[r_t^{\mathrm{G}}(i)] = \mu^{\mathrm{G}}(i)$$

In our protocol when the learner opts to observe a ground truth measurement, she receives a two-dimensional tuple $[r_t^{\mathrm{G}}(i_t), r_t^{\mathrm{C}}(i_t)]$. Otherwise she receives a single coarse reward value $r_t^{\mathrm{C}}(i_t)$. We remark that the ground truth and coarse means are not necessarily related to one another, and may in fact encode different orderings of arms. This is a setting which we call *mis-alignment*: the coarse means implicitly encode different reward preferences than the ground truth. We do however, assume that the two measurements are correlated. We denote the *unknown* variance of the coarse and ground truth data $[r^{\mathrm{G}}(i),\ r^{\mathrm{C}}(i)]$ as well as their *unknown* covariance as $\mathrm{Var}^{\mathrm{C}}(i), \mathrm{Var}^{\mathrm{G}}(i)$, and $\mathrm{Cov}(i)$ respectively.

A nonzero covariance between measurements (given an arm) ensures that there is statistical signal to extract from coarse data which is informative for explaining away some of the variation in the ground truth measurements. The correlation can be quantified by the correlation coefficient: $\rho(i) = \frac{\mathrm{Cov}(i)}{\sqrt{\mathrm{Var}^{\mathrm{G}}(i)\mathrm{Var}^{\mathrm{C}}(i)}} \in [-1, 1]$. Given these definitions, we make two key assumptions on these random variables. The first is a common boundedness assumption.

**Assumption 4.1.** *There exists a* known *value $b > 0$ such that $r^{\mathrm{C}}(i), r^{\mathrm{G}}(i) \in [0, b]$ almost surely, for all $i \in [K]$.*

The second assumption is of knowledge of an upper bound on the ratio of $\mathrm{Var}^{\mathrm{G}}$ and $\mathrm{Var}^{\mathrm{C}}$.

**Assumption 4.2.** *There exists a known $\gamma > 0$, such that $\mathrm{Var}^{\mathrm{G}}(i) \leqslant \gamma^2 \mathrm{Var}^{\mathrm{C}}(i)$ for all $i \in [K]$.*

This assumption is also natural, since when $\gamma = 1$, this parameter simply reflects the (natural) setting in which the coarse measurements are "more stochastic" measurements of the ground truth.

Given this setup, we assume the learner seeks to minimize their expected regret when compared to the optimal (ground-truth) arm.

$$\mathrm{Regret}(T) = \sum_{t=1}^{T} \max_{i \in [K]} \mu^{\mathrm{G}}(i) - \mu^{\mathrm{G}}(i_t)$$

Running a no-regret algorithm such as UCB (Lai & Robbins, 1985) and setting the measurement type to ground truth at every time step (i.e. $m_t = \mathrm{G}$ for all $t \in \mathbb{N}$) achieves an expected regret bound of order $\mathcal{O}(\sqrt{KT \log T})$. Unfortunately, this would incur a cost of $T$ expensive ground measurements. In our problem, however, ground truth measurements are comparatively expensive to collect, and the user attempts to limit their usage to a user-specified ratio $\alpha(i)$ of pulls, where $\alpha(i) = 1$ corresponds to an algorithm using only ground truth measurements, and $\alpha(i) = 0$ an algorithm using only coarse measurements.

We define $\mathcal{T}_t^{\mathrm{G}}(i)$ as the set of time-steps $\ell \in [t]$ where $m_\ell = \mathrm{G}$ and $i_\ell = i$ similarly $\mathcal{T}_t^{\mathrm{C}}(i)$ is the set of time-steps $\ell \in [t]$ where $m_\ell = \mathrm{C}$ and $i_\ell = i$. With this notation, we can represent the number of ground and coarse pulls for arm $i$ at time $t$ as $N_t^{\mathrm{G}}(i) = |\mathcal{T}_t^{\mathrm{G}}(i)|$ and $N_t^{\mathrm{C}}(i) = |\mathcal{T}_t^{\mathrm{C}}(i)|$. The total number of pulls for arm $i$ is denoted $N_t(i) = N_t^{\mathrm{G}}(i) + N_t^{\mathrm{C}}(i)$.

To keep the actual ratio $\alpha_t(i) := \frac{N_t^{\mathrm{G}}(i)}{N_t(i)}$ close to the desired $\alpha(i) \in (0,1)$, we develop a simple thresholding rule:

$$
m_t = \begin{cases} \mathrm{G} & : & N_{t-1}^{\mathrm{G}}(i_t) \leqslant \alpha(i) N_{t-1}(i_t) \\ \mathrm{C} & : & \text{otherwise} \end{cases} . \tag{2}
$$

For convenience, we define the ratio $x_t(i) := \frac{N_t^{\mathrm{G}}(i)}{N_t^{\mathrm{C}}(i)} = \frac{\alpha_t(i)}{1-\alpha_t(i)}$, i.e., the ratio of ground to coarse pulls for the arm $i$. Then the target value of $x_t(i)$ is given by $x(i) := \frac{\alpha(i)}{1-\alpha(i)}$. The following result guarantees that this decision rule controls $\alpha_t(i)$ around $\alpha(i)$. The proof is provided in Appendix B.

**Lemma 4.1** (Feedback ratio guarantee)**.** *The threshold rule* (2) *guarantees the following bounds on the ratio $\alpha_t(i)$ for each arm $i \in [K]$ and all times $t$:*

$$
\alpha(i) \left( 1 - \frac{1}{N_t(i)} \right) \leqslant \alpha_t(i) \leqslant \alpha(i) \left( 1 - \frac{1}{N_t(i)} \right) + \frac{1}{N_t(i)}
$$

In the next section we combine this rule with statistical estimators for the ground truth means to derive a new multi-armed bandit algorithm: Coupled Uncertainty Upper Confidence Bounds (CUUCB).

## 5 COUPLED UNCERTAINTY UPPER CONFIDENCE BOUNDS

In this section we define and analyze the statistical estimators used in the CUUCB algorithm. These estimators exploit the coupling between the ground truth and coarse data and have tighter confidence intervals which scale with the quantity $C_\alpha$ (see (1)) as opposed to the quantity $C'_\alpha$ which would be possible using ground truth measurements alone. As we show, using these estimators in an upper confidence bounds algorithm results in lower regret on the multi-armed bandit problem.

### 5.1 UNBIASED AND REDUCED VARIANCE ESTIMATORS OF THE GROUND TRUTH MEAN

To begin constructing our estimators, recall that at time $t$, our algorithm CUUCB pulls arm $i_t$ and then selects a measurement type $m_t \in \{\mathrm{C}, \mathrm{G}\}$ after which it observes feedback in the form of either a coarse measurement from arm $i_t$ or a tuple of coarse/ground-truth measurements. Given this feedback, we define empirical estimators of $\mu^{\mathrm{G}}(i)$ and $\mu^{\mathrm{C}}(i)$ at time $t$ given by

$$
\mu_t^X(i) := \frac{1}{N_t^X(i)} \sum_{j \in \mathcal{T}_t^X(i)} r_j^X(i), \ X = \mathrm{G}, \mathrm{C}.
$$

We also define the estimator for $\mu^{\mathrm{C}}(i)$ which is derived solely from data collected from coupled measurements $\mu_t^{\mathrm{C},\mathrm{G}}(i)$:

$$
\mu_t^{\mathrm{C},\mathrm{G}}(i) := \frac{1}{N_t^{\mathrm{G}}(i)} \sum_{j \in \mathcal{T}_t^{\mathrm{G}}(i)} r_j^{\mathrm{C}}(i).
$$

Our estimator for the ground truth mean $\mu^{\mathrm{G}}(i)$, linearly combines $\mu_t^{\mathrm{G}}(i)$, $\mu_t^{\mathrm{C},\mathrm{G}}(i)$ and $\mu_t^{\mathrm{C}}(i)$ to for variance reduction as follows:

$$
Z_t^\lambda(i) := \mu_t^{\mathrm{G}}(i) - \lambda(\mu_t^{\mathrm{C},\mathrm{G}}(i) - \mu_t^{\mathrm{C}}(i)),
$$

for some constant $\lambda \in \mathbb{R}$. Note that $Z_t^\lambda(i)$ is an unbiased estimator of $\mu^{\mathrm{G}}(i)$ for any fixed $\lambda$. This estimator is simply the augmented inverse propensity weighted (AIPW) estimator first studied in Robins & Rotnitzky (1995) and more recently in Angelopoulos et al. (2024) which studied the tuning of $\lambda$ to reduce variance and thus the width of asymptotic confidence intervals. We derive finite-sample confidence intervals in the case of unknown variances and correlations.

---

**Algorithm 1** Coupled Uncertainty Upper Confidence Bounds (CUUCB)

---

1: **Input:** $\delta \in (0,1), \alpha \in (0,1)^K, T$.
2: Initialize estimators $\{\mu_0^X(i), \text{Var}_0^X(i), \text{Cov}_0(i)\}_{i \in [K], X \in \{G,C,(C,G)\}}$
3: Initialize $N_0^G(i) = N_0^C(i) = 0$ for $i \in [K]$.
4: Initialize UCB bounds $U_0(i) = b$ for $i \in [K]$.
5: **for** $t = 1, 2, \cdots, T$ **do**
6:     Select arm $i_t = \arg\max_{i \in [K]} U_{t-1}(i)$.
7:     Select feedback according to rule (2).
8:     Update count $N_t^{m_t}(i_t) = N_{t-1}^{m_t}(i_t) + 1$
9:     Update estimators $\mu_t^G(i_t), \mu_t^{C,G}(i_t), \text{Var}_t^G(i_t), \text{Var}_t^{C,G}(i_t), \mu_t^C(i_t), \text{Var}_t^C(i_t)$.
10:    Compute $Z_t(i_t), \hat{V}_t^{\lambda_t(i_t)}(i_t)$ and UCB bound $U_t(i_t)$.
11: **end for**

---

The main idea is to exploit the coupling between the ground and coarse rewards to choose $\lambda$ resulting in a $Z_t^\lambda$ having a lower variance as compared to just using $\mu_t^G$ as the mean estimator. To see that this is possible, we use the fact that $(\mu_t^G(i) - \lambda \mu_t^C(i))$ and $\lambda \hat{\mu}_t^C(i)$ are independent to obtain the expression $\text{Var}(Z_t^\lambda(i)) = \frac{V_t^\lambda(i)}{N_t^G(i)}$, where $V_t^\lambda(i)$ is the following quadratic expression in $\lambda$:

$$V_t^\lambda(i) = (1 + x_t(i))\text{Var}^C(i)\lambda^2 - 2\text{Cov}(i)\lambda + \text{Var}^G(i).$$

One then observes that the choice of $\lambda$ that minimizes $V_t^\lambda(i)$ is given by $\lambda_t^*(i) = \frac{(1-\alpha_t(i))\text{Cov}(i)}{\text{Var}^C(i)}$, which results in $Z_t^\lambda(i)$ having variance $\frac{V_t^*(i)}{N_t^G(i)}$, where

$$V_t^*(i) = \text{Var}^G(i) - \frac{(1-\alpha_t(i))\text{Cov}(i)^2}{\text{Var}^C(i)} = \text{Var}^G(i)\left(1 - (1-\alpha_t(i))\rho(i)^2\right).$$

Note that by assumption 4.2, we have $|\lambda_t^*(i)| \leqslant (1 - \alpha_t(i))\gamma$. We remark that $V_t^*(i) \leqslant \text{Var}^G(i)$ for all values of $\alpha$ and $\rho$ which means variance reduction is obtained by using the estimator $Z_t^{\lambda_t^*(i)}(i)$ in place of $\mu_t^G(i)$. Through Bennett's concentration inequality we immediately observe how one can exploit this variance reduction to achieve an improved upper confidence bound for $\mu^G(i)$.

**Lemma 5.1** (Bennett's Inequality). *Given an arm $i$ and fixing the number of its pulls $N_t^C(i), N_t^G(i)$, consider a fixed $\lambda \in \mathbb{R}$ satisfying $|\lambda| \leqslant (1 - \alpha_t(i))\gamma$. Then the following concentration inequality holds with probability at least $1 - \delta$:*

$$|Z_t^\lambda(i) - \mu^G(i)| \leqslant \sqrt{\frac{2V_t^\lambda(i)\log\left(\frac{2}{\delta}\right)}{N_t^G(i)}} + \frac{(1+\gamma)b\log\left(\frac{2}{\delta}\right)}{3N_t^G(i)}.$$

The proof of this follows from a simple application of Bennett's concentration inequality and is supplied in Appendix B.1 for completeness.. Lemma 5.1 gives us a concentration radius with the leading term dependent on $V_t^\lambda(i)$. Thus, picking $\lambda = \lambda_t^*(i)$ would allow us to obtain the tightest concentration and thus a lower regret algorithm. However, we do not know $\Sigma(i)$ and thus cannot directly compute $V_t^\lambda(i)$ or $\lambda_t^*(i)$. As we show, one can derive a version of this bound using empirical estimators of these quantities.

## 5.2 EMPIRICAL ESTIMATION OF VARIANCES AND COVARIANCES

To fully exploit our AIPW estimator without knowledge of the variances or correlations between ground truth and coarse measurements, we introduce empirical estimators of

$\text{Var}^{\text{G}}(i), \text{Var}^{\text{C}}(i), \text{Var}_t^{\text{C,G}}(i)$ and $\text{Cov}(i)$ at time $t$ as:

$$\text{Var}_t^X(i) := \frac{1}{N_t^X(i)} \sum_{j \in \mathcal{T}_t^X(i)} (r_j^X(i) - \mu_t^X(i))^2, \; X = \text{G}, \text{C}$$

$$\text{Var}_t^{\text{C,G}}(i) := \frac{1}{N_t^{\text{G}}(i)} \sum_{j \in \mathcal{T}_t^{\text{G}}(i)} (r_j^{\text{C}}(i) - \mu_t^{\text{C,G}}(i))^2$$

$$\text{Cov}_t(i) := \frac{1}{N_t^{\text{G}}(i)} \sum_{j \in \mathcal{T}_t^{\text{G}}(i)} (r_j^{\text{G}}(i) - \mu_t^{\text{G}}(i))(r_j^{\text{C}}(i) - \mu_t^{\text{C}}(i)).$$

Note that we can only compute the covariance estimator due to the coupling between the two sources of feedback given an outcome. These definitions allow us to define the following empirical estimator for $\text{V}_t^\lambda(i)$ as a function of $\lambda$:

$$\hat{\text{V}}_t^\lambda(i) = (1 + x_t(i))\widehat{\text{Var}}_t^{\text{C}}(i)\lambda^2 - 2\text{Cov}_t(i)\lambda + \text{Var}_t^{\text{G}}(i),$$

where for convenience we define $\widehat{\text{Var}}_t^{\text{C}}(i) := (1 - \alpha_t(i))\text{Var}_t^{\text{C,G}}(i) + \alpha_t(i)\text{Var}_t^{\text{C}}(i)$. The expression for $\hat{\text{V}}_t^\lambda$ can be obtained by simply replacing the variances and covariance in the expression for $\text{V}_t^\lambda$ with their appropriate empirical estimators. We analyze its concentration around $V_t^\lambda(i)$ in Appendix B.2. We also define an empirical estimator of $\lambda_t^*(i)$:

$$\lambda_t(i) := \underset{\lambda \in [-(1-\alpha_t(i))\gamma, (1-\alpha_t(i))\gamma]}{\arg\min} \hat{\text{V}}_t^\lambda(i)$$

$$= (1 - \alpha_t(i))\mathcal{P}_\gamma\left(\frac{\text{Cov}_t(i)}{\widehat{\text{Var}}_t^{\text{C}}(i)}\right) \quad \text{where: } \mathcal{P}_\gamma(x) := \begin{cases} \gamma, & \text{if } x > \gamma \\ x, & \text{if } x \in [-\gamma, \gamma] \\ -\gamma, & \text{if } x < -\gamma \end{cases}.$$

Note that the bounds on $\gamma$ in the minimization follow from assumption 4.2. Plugging-in the estimator $\lambda_t(i)$ into $Z_t^\lambda(i)$, we obtain the following estimator for $\mu_t^{\text{G}}(i)$:

$$Z_t(i) := Z_t^{\lambda_t(i)}(i) = \mu_t^{\text{G}}(i) - \lambda_t(i)(\mu_t^{\text{C,G}}(i) - \mu_t^{\text{C}}(i)).$$

The following lemma, which is an empirical analogue to Lemma 5.1 gives us the concentration of $Z_t(i)$ around $\mu^{\text{G}}(i)$ with a confidence bound that scales with the empirical variance.

**Lemma 5.2** (Anytime concentration of $Z_t(i)$)**.** *Consider a fixed arm $i \in [K]$ and some $\delta \in (0, 1)$. Define $L_t(i) := \log\left(16\delta^{-1}(N_t(i))^2(N_t(i) + 2)\right)$. Then w.p. at least $1 - \delta$, the following inequalities hold for all times $t$:*

$$|Z_t(i) - \mu^{\text{G}}(i)| \leqslant \sqrt{\frac{2\hat{\text{V}}_t^{\lambda_t(i)}(i)L_t(i)}{N_t^{\text{G}}(i)}} + \frac{6(1+\gamma)bL_t(i)}{N_t^{\text{G}}(i)} \leqslant \sqrt{\frac{2\text{V}_t^*(i)L_t(i)}{N_t^{\text{G}}(i)}} + \frac{7(1+\gamma)bL_t(i)}{N_t^{\text{G}}(i)}.$$

The proof of Lemma 5.2 is deferred to Appendix B.2. The proof follows from showing the concentration of the empirical variance and covariance estimators and a careful covering argument to handle the fact that $\lambda_t(i)$ is itself a stochastic quantity.

We use Lemma 5.2 to define an Upper Confidence Bound on the true mean $\mu^{\text{G}}(i)$ of arm $i$ at time $t$:

$$U_t(i) := Z_t(i) + \sqrt{\frac{2\hat{\text{V}}_t^{\lambda_t(i)}(i)L_t(i)}{N_t^{\text{G}}(i)}} + \frac{6(1+\gamma)bL_t(i)}{N_t^{\text{G}}(i)} \tag{3}$$

Using this confidence interval in conjunction with the thresholding rule for maintaining the desired ratio $\alpha(i)$ of ground truth to coarse feedback results in the CUUCB algorithm detailed in Algorithm 1.

## 5.3 REGRET ANALYSIS OF CUUCB

Given our the estimator and its confidence bounds, we now present instance-dependent and instance-independent expected regret bounds for CUUCB, the proof of which is deferred to Appendix B.4 and follows from our concentration guarantees for our variance reduced estimator.

**Theorem 5.3** (Regret Bounds). *Let $\delta = \frac{1}{T}$. Then, the expected regret of CUUCB satisfies the instance-dependent bound:*

$$\mathbb{E}[\text{Regret}(T)] \leqslant 128 \sum_{i:\Delta_i > 0} \frac{1}{\alpha(i)} \left( \frac{\text{V}^*(i)}{\Delta_i} + b(1+\gamma) \right) \log(2T) + \mathcal{O}(1),$$

*where $\Delta_i := \max_{j \in [K]} \mu^{\text{G}}(j) - \mu^{\text{G}}(i)$ is the loss of playing arm $i$ and $\text{V}^*(i) := \text{Var}^{\text{G}}(i) - \frac{(1-\alpha(i))\text{Cov}(i)^2}{\text{Var}^{\text{C}}(i)}$. Moreover, the expected regret also satisfies the instance-independent bound:*

$$\mathbb{E}[\text{Regret}(T)] \leqslant 23 \sqrt{\left( \sum_i \frac{\text{V}^*(i)}{\alpha(i)} \right) T \log(2T)} + \mathcal{O}(\log(T))$$

In the above result, one can think of $\text{V}^*(i)$ as the limiting value of $\text{V}_t^*(i)$ as the number of samples grows large, resulting in the ratio $\alpha_t(i)$ going to $\alpha(i)$. Lemma 5.3 gives us an instance-independent bound on the regret of order $\mathcal{O}(\sqrt{C_\alpha T \log T})$, where

$$C_\alpha = \sum_i \frac{\text{V}^*(i)}{\alpha(i)} = \sum_i \text{Var}^{\text{G}}(i) \left( \frac{(1-\rho(i)^2)}{\alpha(i)} + \rho(i)^2 \right)$$

On the other hand, if we were to use a classical variance adaptive algorithm like UCB-V, which has access to only the $\alpha(i)$ fraction of ground truth measurements but has to play for the same $T$ steps, then the expected regret bound will have the order $\mathcal{O}(\sqrt{C'_\alpha T \log T})$ with $C'_\alpha = \sum_i \frac{\text{Var}^{\text{G}}(i)}{\alpha(i)}$ as shown in Lemma B.3. Thus, we see that $C_\alpha$ has a reduction compared to $C'_\alpha$ due to the $\rho(i)^2$ term. We see that the reduction obtained is greater if $|\rho(i)|$ becomes larger and $\alpha(i)$ becomes smaller, i.e., as the ground and coarse samples become more correlated and as ground samples become more scarce. Thus, CUUCB potentially allows us to exploit the additional coarse samples to obtain a reduction in regret. We look at this behaviour in more detail in the experiments that follow.

## 6 EXPERIMENTS

We now evaluate our algorithms and theory on synthetic data and an LLM-benchmarking task. For details on implementation, dataset generation and additional experimental results, see Appendix C.

### 6.1 SYNTHETIC EXPERIMENTS

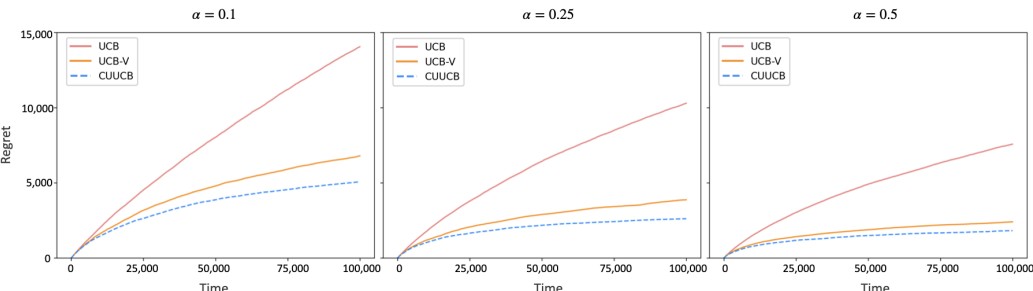

Figure 1: Comparison of CUUCB, UCB and UCB-V on synthetically generated data for fixed value of $\rho = 0.8$ and different values of $\alpha$ with a time horizon of $100,000$ steps, averaged across 40 runs.

In our first experiment, we test CUUCB on synthetically generated data and compare it to UCB and UCB-V. We keep the number of arms as $K = 4$ and generate the coupled ground and coarse data points for all arms $i$ by sampling from a common 2-D jointly Gaussian distribution with correlation set to a parameter $\rho$. We then clip the values in a desired interval to ensure the data distribution has finite support. We choose a uniform target ratio $\alpha(i) = \alpha$ for all arms $i$. The algorithms UCB and UCB-V make decisions using only the ground truth samples observed at a given time $t$, i.e., ground truth samples from time steps $\mathcal{T}_t^{\text{G}}(i)$. Per our theory, we expect CUUCB to outperform the

other algorithms if we pick the correlation $\rho$ to be large and set $\alpha$ to be smaller. We compare the algorithms when $\rho = 0.8$ is fixed and we vary $\alpha$ as seen in Figure 1. We run the experiments for a time horizon $T = 100,000$ and for each algorithm, and show the average regret across 40 runs. We observe that while increasing $\alpha(i)$ leads to an overall reduction in regret for all algorithms, the performance improvement obtained by CUUCB over UCB-V is largest at smaller $\alpha$. Additional results in the setting when $\alpha$ is fixed and $\rho$ is varied can be found in Appendix C.4.1.

## 6.2 LLM-as-a-Judge Experiments

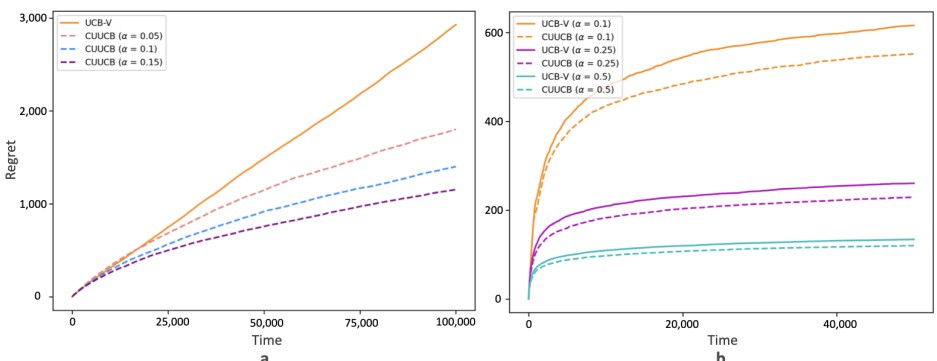

Figure 2: LLM benchmarking regret plots: a. Performance on misaligned Dataset-A of UCB-V which makes decisions using only coarse measurements vs. CUUCB for different $\alpha$ with $T = 100,000$, averaged across 40 runs. b. Performance on Dataset-B of UCB-V which has access to only ground truth samples vs. CUUCB for different $\alpha$ with $T = 50,000$, averaged across 40 runs.

In our second experiment, we study a task of trying to find the best language model (LLM) amongst a set of large language models for a given set of prompts. Since the idea of a "best" model can be highly subjective, this task is normally accomplished by asking people to rank or score (prompt, answer) pairs from each LLM to find the best model Chiang et al. (2024). This approach requires potentially massive amounts of human annotations which can be too expensive to try in practice. To overcome this problem, one emerging approach to LLM benchmarking is to simply ask a holdout LLM to supply the scores—an emerging form of learning from AI feedback Zhu et al. (2024) also increasingly called the LLM-as-a-judge paradigm Gu et al. (2025). Crucially—and as we show in our experiments— one cannot simply rely on *only* AI feedback as the AI feedback can be misaligned and fail to correctly capture human preferences. As such, a combination of human and AI feedback is required to accurately benchmark LLMs to human preferences.

This problem fits neatly into our framework: the human annotations take the place of our ground-truth measurements against which the algorithm's performance will ultimately be benchmarked, while large language model annotations become the coarse measurement.

To simulate this problem we make use of the Nectar dataset of rankings of (prompt,answer) pairs from Zhu et al. (2024). This dataset comprises of 182,954 prompts, each answered by seven LLMs. These answers are then scored by GPT-4 Bubeck et al. (2023). To design this experiment, we extract (prompt, answer) pairs for six chosen models from this dataset and use a pair of LLMs to score the models. We treat the scores given by the more advanced LLM as the ground-truth (i.e., proxies for human labels) and those by the less advanced LLM as coarse feedback. By querying different LLMs, we obtain two datasets–Dataset-A and Dataset-B, consisting of ground-truth and coarse scores of the six models. The dataset generation details can be found in Appendix C.3.

Given this setup, our experiment proceeds in two parts. First, we observe that the ground-truth and coarse scores in Dataset-A are misaligned, i.e., the model with the best mean score differs for the two models. Thus, an attempt to use a MAB algorithm like UCB-V with only coarse feedback leads to linear regret when measured using the ground-truth means. We observe that CUUCB is able to overcome the misalignment while using only a small fraction $\alpha$ of ground-truth samples, as shown in Figure 2a. In the second part, we compare regrets of CUUCB and UCB-V on Dataset-B while varying the parameter $\alpha$ as shown in Figure 2b. We consistently observe that our algorithm outperforms and that the advantage is largest when the amount of ground truth data is most restricted.

## REPRODUCIBILITY STATEMENT

To ensure reproducibility of our theoretical results, detailed proofs are provided in Appendix A and Appendix B. To ensure our experimental results are reproducible, we include experimental details such as algorithm implementations and dataset generation in Appendix C. Additionally, we have submitted all of our code and important datasets as part of the supplementary materials.

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

## A  SUPPORTING TECHNICAL RESULTS

**Lemma A.1.** *[Adapted from Audibert] Let $\delta \in (0, 1)$ and $Z_1, \cdots, Z_n$ be i.i.d. random variables with values in $[0, b]$. Define $\bar{Z}_n = \frac{1}{n} \sum_{i=1}^{n} Z_i$, $\mathrm{Var}_n = \frac{1}{n} \sum_{i=1}^{n} (Z_i - \bar{Z}_n)^2$ and $\mathrm{Var} = \mathbb{E}[(Z - \mathbb{E}[Z])^2]$. With probability at least $1 - \delta$,*

$$\sqrt{\mathrm{Var}_n} \leqslant \sqrt{\mathrm{Var}} + b\sqrt{\frac{\log(1/\delta)}{2n}}.$$

*We also have with probability at least $1 - \delta$ that:*

$$\sqrt{\mathrm{Var}} \leqslant \sqrt{\mathrm{Var}_n} + 1.8b\sqrt{\frac{\log(3/\delta)}{n}}$$

*Proof.* First, we prove the high probability upper bound. For this, we note that we can bound:

$$\mathrm{Var}_n = \frac{1}{n} \sum_{i=1}^{n} (Z_i - \bar{Z}_n)^2 = \frac{1}{n} \sum_{i=1}^{n} (Z_i - \mathbb{E}[Z])^2 - (\bar{Z}_n - \mathbb{E}[Z])^2$$

$$\leqslant \frac{1}{n} \sum_{i=1}^{n} (Z_i - \mathbb{E}[Z])^2 \tag{4}$$

Noting that $\{(Z_i - \mathbb{E}[Z])^2\}_{i=1}^{n}$ are i.i.d. random variables lying almost surely in $[0, b^2]$, we use Bennet's inequality to obtain with probability at least $1 - \delta$ that

$$\frac{1}{n} \sum_{i=1}^{n} (Z_i - \mathbb{E}[Z])^2 - \mathrm{Var} \leqslant \sqrt{\frac{2\mathcal{V}\log(1/\delta)}{n}} + \frac{b^2\log(1/\delta)}{3n}, \tag{5}$$

where $\mathcal{V} = \mathbb{E}[(Z - \mathbb{E}[Z])^4] - \text{Var}^2 \leqslant b^2 \mathbb{E}[(Z - \mathbb{E}[Z])^2] = b^2 \text{Var}$. Substituting this bound in (5), and combining with (4).], we obtain:

$$\text{Var}_n \leqslant \text{Var} + \sqrt{\frac{2b^2 \text{Var} \log(1/\delta)}{n}} + \frac{b^2 \log(1/\delta)}{3n}$$

$$\leqslant \text{Var} + \sqrt{\frac{2b^2 \text{Var} \log(1/\delta)}{n}} + \frac{b^2 \log(1/\delta)}{2n} = \left( \sqrt{\text{Var}} + b\sqrt{\frac{\log(1/\delta)}{2n}} \right)^2$$

Taking square root on both sides above gives us:

$$\sqrt{\text{Var}_n} \leqslant \sqrt{\text{Var}} + b\sqrt{\frac{\log(1/\delta)}{2n}}$$

which proves the upper bound. The lower bound is taken from a result in the appendix of Audibert et al. (2009). □

## B  PROOFS FOR COUPLED UNCERTAINTY UCB

To begin we prove the fact that the thresholding rule (2) maintains the ratio of coarse to ground truth feedback to within the desired range.

*Proof of Lemma 4.1.* We drop the index $i$ for brevity. At time $t$ let us consider the following:

1. Upper bound: Let us consider the latest time $t_1 < t$ when the decision to pull G was taken. Then we must have had $N_t^G - 1 = N_{t_1}^G \leqslant xN_{t_1}^C \leqslant xN_t^C$. On dividing both sides by $N_t$, this gives us:

$$\alpha_t - \frac{1}{N_t} \leqslant x(1 - \alpha_t)$$

$$\Rightarrow \alpha_t \leqslant \frac{x}{(1+x)} + \frac{1}{(1+x)N_t} = \alpha + \frac{1-\alpha}{N_t},$$

which proves the upper bound.

2. Lower bound: Let us similarly consider the latest time $t_2 < t$ when the decision to pull C was taken. Then we must have had $x(N_t^C - 1) = xN_{t_2}^C \leqslant N_{t_2}^G \leqslant N_t^G$. On dividing both sides by $N_t$, this gives us:

$$x\left(1 - \alpha_t - \frac{1}{N_t}\right) \leqslant \alpha_t$$

$$\Rightarrow \alpha_t \geqslant \frac{x}{(1+x)}\left(1 - \frac{1}{N_t}\right) = \alpha\left(1 - \frac{1}{N_t}\right),$$

which proves the lower bound.

□

### B.1  PROOFS FOR THE UNBIASED MEAN ESTIMATOR

In this subsection, we present the proof of the concentration of the mean estimator $Z_t^\lambda$.

*Proof of Lemma 5.1.* We drop the index $i$ for brevity. We apply Bennett's inequality to

$$Z_t^\lambda = \sum_{j \in T_t^G} \frac{(r_j^G - \lambda r_j^C)}{N_t^G} + \sum_{j \in T_t^C} \frac{\lambda r_j^C}{N_t^C}$$

by treating it as the sum of the $N_t$ independent random variables in the collection $\mathcal{H} = \left\{ \frac{(r_j^G - \lambda r_j^C)}{N_t^G} \right\}_{j \in T_t^G} \bigcup \left\{ \frac{\lambda r_j^C}{N_t^C} \right\}_{j \in T_t^C}$.

We use the fact that each random variable in $\mathcal{H}$ has a bounded support of length $\leqslant B$, where

$$B = \frac{b(1 + |\lambda|)}{N_t^{\mathrm{G}}} + \frac{b|\lambda|}{N_t^{\mathrm{C}}} \leqslant \frac{(1 + \gamma)b}{N_t^{\mathrm{G}}},$$

when $|\lambda| \leqslant \frac{\gamma}{(1+x_t)}$. Bennett's inequality then gives us the following concentration result w.p. at least $1 - \delta$:

$$|Z_t^\lambda - \mathbb{E}[Z_t^\lambda]| \leqslant \sqrt{2 \sum_{X \in \mathcal{H}} \mathrm{Var}(X) \log(2/\delta)} + \frac{B \log(2/\delta)}{3} = \sqrt{2\mathrm{Var}(Z_t^\lambda) \log(2/\delta)} + \frac{B \log(2/\delta)}{3}$$

$$\Rightarrow |Z_t^\lambda - \mu^{\mathrm{G}}| \leqslant \sqrt{\frac{2\mathrm{V}_t^\lambda \log(2/\delta)}{N_t^{\mathrm{G}}}} + \frac{(1 + \gamma)b \log(2/\delta)}{3N_t^{\mathrm{G}}}.$$

where we use the fact that $\mathbb{E}[Z_t^\lambda] = \mu^{\mathrm{G}}$. $\qquad\square$

### B.2 PROOFS FOR THE CONCENTRATION OF THE EMPIRICAL VARIANCE ESTIMATORS

In this subsection we present the proofs on the concentration of the empirical variance $\hat{\mathrm{V}}_t^\lambda(i)$.

**Lemma B.1** (Concentration of $\hat{\mathrm{V}}_t^\lambda(i)$)**.** *Given an arm $i$ and fixing the number of its pulls $N_t^{\mathrm{C}}(i), N_t^{\mathrm{G}}(i)$, consider a fixed $\lambda \in \mathbb{R}$ satisfying $|\lambda| \leqslant (1 - \alpha_t(i))\gamma$. Then the following concentration inequalities hold:*

*1. With probability at least $1 - \delta$:*

$$\sqrt{\hat{\mathrm{V}}_t^\lambda(i)} \leqslant \sqrt{\mathrm{V}_t^\lambda(i)} + b(1 + \gamma)\sqrt{\frac{\log(2/\delta)}{2N_t^{\mathrm{G}}(i)}}$$

*2. With probability at least $1 - \delta$:*

$$\sqrt{\mathrm{V}_t^\lambda(i)} \leqslant \sqrt{\hat{\mathrm{V}}_t^\lambda(i)} + 1.8b(1 + \gamma)\sqrt{\frac{\log(6/\delta)}{N_t^{\mathrm{G}}(i)}}$$

*Proof of Lemma B.1.* We omit the index $i$ for brevity. Note that we can split $\hat{\mathrm{V}}_t^\lambda$ into two terms as follows:

$$\frac{\hat{\mathrm{V}}_t^\lambda}{N_t^{\mathrm{G}}} = \frac{\widehat{\mathrm{Var}}_1}{N_t^{\mathrm{G}}} + \frac{\widehat{\mathrm{Var}}_2}{N_t^{\mathrm{C}}},$$

where

$$\widehat{\mathrm{Var}}_1 := \mathrm{Var}_t^{\mathrm{C},\mathrm{G}}\lambda^2 - 2\mathrm{Cov}_t\lambda + \mathrm{Var}_t^{\mathrm{G}} = \frac{1}{N_t^{\mathrm{G}}} \sum_{j \in \mathcal{T}_t^{\mathrm{G}}} (r_j^{\mathrm{G}} - \lambda r_j^{\mathrm{C}})^2$$

$$\widehat{\mathrm{Var}}_2 := \lambda^2 \mathrm{Var}_t^{\mathrm{C}}(i) = \frac{1}{N_t^{\mathrm{C}}} \sum_{j \in \mathcal{T}_t^{\mathrm{C}}} (\lambda r_j^{\mathrm{C}})^2$$

We see that $\widehat{\mathrm{Var}}_1, \widehat{\mathrm{Var}}_2$ are empirical estimators of the variances $\mathrm{Var}_1, \mathrm{Var}_2$ (resp.) as defined below:

$$\mathrm{Var}_1 = \mathrm{Var}(r^{\mathrm{G}} - \lambda r^{\mathrm{C}}) = \mathrm{Var}^{\mathrm{C}}\lambda^2 - 2\mathrm{Cov}\lambda + \mathrm{Var}^{\mathrm{G}}$$

$$\mathrm{Var}_2 = \mathrm{Var}(\lambda r^{\mathrm{C}}) = \lambda^2 \mathrm{Var}^{\mathrm{C}}.$$

Also note that $\frac{\mathrm{V}_t^\lambda}{N_t^{\mathrm{G}}} = \frac{\mathrm{Var}_1}{N_t^{\mathrm{G}}} + \frac{\mathrm{Var}_2}{N_t^{\mathrm{C}}}$.

We now show the first concentration inequality by considering the following two concentration events:

$$\mathcal{E}_1 = \left\{ \sqrt{\widehat{\mathrm{Var}}_1} \leqslant \sqrt{\mathrm{Var}_1} + b(1 + |\lambda|)\sqrt{\frac{\log(4/\delta)}{2N_t^{\mathrm{G}}}} \right\}$$

$$\mathcal{E}_2 = \left\{ \sqrt{\widehat{\mathrm{Var}_2}} \leqslant \sqrt{\mathrm{Var}_2} + b|\lambda| \sqrt{\frac{\log(4/\delta)}{2N_t^{\mathrm{C}}}} \right\}$$

From lemma A.1, it follows that each of the above events occurs w.p. at least $1 - \delta/2$, and so we prove our result on $\mathcal{E}_1 \cap \mathcal{E}_2$ which occurs w.p. at least $1 - \delta$. First for the upper bound, we combine the upper bounds in $\mathcal{E}_1, \mathcal{E}_2$ to get:

$$\frac{\hat{V}_t^\lambda}{N_t^{\mathrm{G}}} = \frac{\widehat{\mathrm{Var}_1}}{N_t^{\mathrm{G}}} + \frac{\widehat{\mathrm{Var}_2}}{N_t^{\mathrm{C}}}$$

$$\leqslant \frac{1}{N_t^{\mathrm{G}}} \left( \mathrm{Var}_1 + 2w_1 \sqrt{\frac{\mathrm{Var}_1}{N_t^{\mathrm{G}}}} + \frac{(w_1)^2}{N_t^{\mathrm{G}}} \right) + \frac{1}{N_t^{\mathrm{C}}} \left( \mathrm{Var}_2 + 2w_2 \sqrt{\frac{\mathrm{Var}_2}{N_t^{\mathrm{C}}}} + \frac{(w_2)^2}{N_t^{\mathrm{C}}} \right)$$

$$= \left( \frac{\mathrm{Var}_1}{N_t^{\mathrm{G}}} + \frac{\mathrm{Var}_2}{N_t^{\mathrm{C}}} \right) + 2 \left( \frac{w_1}{N_t^{\mathrm{G}}} \right) \sqrt{\frac{\mathrm{Var}_1}{N_t^{\mathrm{G}}}} + 2 \left( \frac{w_2}{N_t^{\mathrm{C}}} \right) \sqrt{\frac{\mathrm{Var}_2}{N_t^{\mathrm{C}}}} + \left( \frac{w_1}{N_t^{\mathrm{G}}} \right)^2 + \left( \frac{w_2}{N_t^{\mathrm{C}}} \right)^2$$

$$\leqslant \frac{\mathrm{V}_t^\lambda}{N_t^{\mathrm{G}}} + 2 \left( \frac{w_1}{N_t^{\mathrm{G}}} + \frac{w_2}{N_t^{\mathrm{C}}} \right) \sqrt{\frac{\mathrm{V}_t^\lambda}{N_t^{\mathrm{G}}}} + \left( \frac{w_1}{N_t^{\mathrm{G}}} + \frac{w_2}{N_t^{\mathrm{C}}} \right)^2$$

where $w_1 = 2^{-1/2}b(1 + |\lambda|)\sqrt{\log(4/\delta)}$, $w_2 = 2^{-1/2}b|\lambda|\sqrt{\log(4/\delta)}$. Taking sqaure root and multiplying by $\sqrt{N_t^{\mathrm{G}}}$ on both sides gives us:

$$\sqrt{\hat{\mathrm{V}}_t^\lambda} \leqslant \sqrt{\mathrm{V}_t^\lambda} + \frac{(w_1 + x_t w_2)}{\sqrt{N_t^{\mathrm{G}}}} \leqslant \sqrt{\mathrm{V}_t^\lambda} + b(1 + \gamma)\sqrt{\frac{\log(4/\delta)}{2N_t^{\mathrm{G}}}},$$

where the last inequality above uses the assumption that $(1 + x_t)|\lambda| \leqslant \gamma$. This proves the first concentration result.

The second concentration result follows from repeating the same steps as above by starting with $\frac{V_t^\lambda}{N_t^{\mathrm{G}}} = \frac{\mathrm{Var}_1}{N_t^{\mathrm{G}}} + \frac{\mathrm{Var}_2}{N_t^{\mathrm{C}}}$ and upper bounding each term using the events $\mathcal{E}_1', \mathcal{E}_2'$ as defined below.

$$\mathcal{E}_1' = \left\{ \sqrt{\mathrm{Var}_1} \leqslant \sqrt{\widehat{\mathrm{Var}_1}} + 1.8b(1 + |\lambda|)\sqrt{\frac{\log(6/\delta)}{N_t^{\mathrm{G}}}} \right\}$$

$$\mathcal{E}_2' = \left\{ \sqrt{\mathrm{Var}_2} \leqslant \sqrt{\widehat{\mathrm{Var}_2}} + 1.8b|\lambda|\sqrt{\frac{\log(6/\delta)}{N_t^{\mathrm{C}}}} \right\}$$

$\mathcal{E}_1', \mathcal{E}_2'$ each holds w.p. at least $1 - \delta/2$ from Lemma A.1, so the concentration result holds on their intersection w.p. at least $1 - \delta$. $\qquad\square$

### B.3 Constructing the Coupled Uncertainty Upper Confidence Bound

This section develops the final proofs for the upper confidence bound used in CUUCB.

**Lemma B.2** (Variance adaptive concentration of $Z_t^\lambda(i)$). *Given an arm $i$ and fixing the number of its pulls $N_t^{\mathrm{C}}(i), N_t^{\mathrm{G}}(i)$, consider a fixed $\lambda \in \mathbb{R}$ satisfying $|\lambda| \leqslant (1 - \alpha_t(i))\gamma$. Then the following concentration inequality holds with probability at least $1 - \delta$:*

$$|Z_t^\lambda(i) - \mu^{\mathrm{G}}(i)| \leqslant \sqrt{\frac{2\hat{V}_t^\lambda(i) \log\left(\frac{8}{\delta}\right)}{N_t^{\mathrm{G}}(i)}} + \frac{3(1 + \gamma)b\log\left(\frac{8}{\delta}\right)}{N_t^{\mathrm{G}}(i)}$$

*Proof of Lemma B.2.* We omit the index $i$ for brevity. Consider the following concentration events.

$$\mathcal{E}_1 = \left\{ |Z_t^\lambda - \mu^{\mathrm{G}}| \leqslant \sqrt{\frac{2\mathrm{V}_t^\lambda \log(8/\delta)}{N_t^{\mathrm{G}}}} + \frac{(1 + \gamma)b\log(8/\delta)}{3N_t^{\mathrm{G}}} \right\}$$

$$\mathcal{E}_{2,t} = \left\{ \sqrt{\mathrm{V}_t^\lambda} \leqslant \sqrt{\hat{V}_t^\lambda} + 1.8b(1 + \gamma)\sqrt{\frac{\log(8/\delta)}{N_t^{\mathrm{G}}}} \right\}$$

From lemma 5.1, we have that $\mathbb{P}(\mathcal{E}_1^c) \leqslant \frac{\delta}{4}$. From lemma B.1, we have that $\mathbb{P}(\mathcal{E}_2^c) \leqslant \frac{3\delta}{4}$. We thus show the concentration result on the event $\mathcal{E}_1 \cap \mathcal{E}_2$ which holds w.p. at least $1 - \delta$.

We denote $l_\delta = \log(8/\delta)$ for simplicity. Combining the two inequalities, we have:

$$|Z_t^\lambda - \mu^{\mathrm{G}}| \leqslant \sqrt{\frac{2\mathrm{V}_t^\lambda l_\delta}{N_t^{\mathrm{G}}}} + \frac{(1+\gamma)bl_\delta}{3N_t^{\mathrm{G}}}$$

$$\leqslant \sqrt{\frac{2\hat{\mathrm{V}}_t^\lambda l_\delta}{N_t^{\mathrm{G}}}} + \frac{1.8\sqrt{2}(1+\gamma)bl_\delta}{N_t^{\mathrm{G}}} + \frac{(1+\gamma)bl_\delta}{3N_t^{\mathrm{G}}} \leqslant \sqrt{\frac{2\hat{\mathrm{V}}_t^\lambda l_\delta}{N_t^{\mathrm{G}}}} + \frac{3(1+\gamma)bl_\delta}{N_t^{\mathrm{G}}},$$

which proves the result. $\qquad\square$

*Proof of Lemma 5.2.* Let us first fix the number of pulls $N_t(i)$ of the arm $i$. Note that because of our deterministic decision rule for $m_t$, $N_t^{\mathrm{G}}(i), N_t^{\mathrm{C}}(i)$ are also fixed upon fixing $N_t(i)$. Also, all the estimators for arm $i$ are evaluated using the outcomes of these $N_t(i)$ pulls. In the following steps, we omit the index $i$ for brevity.

Let us define $\gamma' := (1 - \alpha_t)\gamma$. Note that our estimator $\lambda_t \in [-\gamma', \gamma']$ by construction. Thus, we define a grid on the interval $[-\gamma', \gamma']$ as follows: Define $\epsilon := \frac{2\gamma'}{N_t}$. We cover the interval $[-\gamma, \gamma]$ using $N_t$ $\epsilon$-width intervals $\{I_j := [\beta_{j-1}, \beta_j]\}_{j=1}^{N_t}$, where $\beta_j := -\gamma' + j\epsilon$.

We define $\delta_n := \frac{\delta}{2n^2}$. Note that $\sum_{n \geqslant 1} \delta_n = \frac{\delta}{2} \sum_{n \geqslant 1} \frac{1}{n^2} \leqslant \delta$.

We have $L_t = \log\left(\frac{8(N_t+2)}{\delta_{N_t}}\right)$. Now, let us consider the following concentration events.

$$\mathcal{E}_{1,N_t} = \left\{ |Z_t^\lambda - \mu^{\mathrm{G}}| \leqslant \sqrt{\frac{2\hat{\mathrm{V}}_t^\lambda L_t}{N_t^{\mathrm{G}}}} + \frac{3(1+\gamma)bL_t}{N_t^{\mathrm{G}}}, \forall \lambda \in \{\beta_0, \beta_1, ..., \beta_{N_t}\} \right\}$$

$$\mathcal{E}_{2,N_t} = \left\{ \sqrt{\hat{\mathrm{V}}_t^{\lambda^*}} \leqslant \sqrt{\mathrm{V}_t^{\lambda^*}} + b(1+\gamma)\sqrt{\frac{L_t}{2N_t^{\mathrm{G}}}} \right\}$$

From lemma B.2 and a union bound, we have that $\mathbb{P}(\mathcal{E}_{1,N_t}^c) \leqslant \frac{(N_t+1)\delta_{N_t}}{(N_t+2)}$. From lemma B.1 and a union bound, we have that $\mathbb{P}(\mathcal{E}_{2,N_t}^c) \leqslant \frac{\delta_{N_t}}{4(N_t+2)} \leqslant \frac{\delta_{N_t}}{(N_t+2)}$. Now, we consider the intersection event $\mathcal{E}_{N_t} := \bigcap_{i=1}^2 \mathcal{E}_{i,N_t}$. Then a union bound gives us $\mathbb{P}(\mathcal{E}_{N_t}^c) \leqslant \delta_{N_t}$. We prove our bounds on the event $\mathcal{E}_{N_t}$ which occurs w.p. at least $1 - \delta_{N_t}$.

First, we derive some useful inequalities. Given any $\lambda, \lambda' \in [-\gamma', \gamma']$ satisfying $|\lambda - \lambda'| \leqslant \epsilon$, we have:

$$|\hat{\mathrm{V}}_t^\lambda - \hat{\mathrm{V}}_t^{\lambda'}| \leqslant (1 + x_t)\widehat{\mathrm{Var}}_t^{\mathrm{C}}(|\lambda| + |\lambda'|)|\lambda - \lambda'| + 2|\lambda - \lambda'||\mathrm{Cov}_t|$$
$$\overset{(i)}{\leqslant} 2(\gamma\widehat{\mathrm{Var}}_t^{\mathrm{C}} + |\mathrm{Cov}_t|)\epsilon \overset{(ii)}{\leqslant} \frac{1}{N_t}\gamma(\gamma + 4)b^2, \tag{6}$$

where in step $(i)$ we have used the fact that $|\lambda|, |\lambda'| \leqslant \gamma'$. In step $(ii)$ we have used the fact that $\widehat{\mathrm{Var}}^{\mathrm{C}} \leqslant \frac{b^2}{4}, |\mathrm{Cov}_t| \leqslant b^2$.

Now, consider an arbitrary $\lambda \in [-\gamma', \gamma']$. We derive an upper bound on $|Z_t^\lambda - \mu^{\mathrm{G}}|$. For this, suppose $j \in [N_t]$ is such that $\lambda \in I_j$. Then for $\lambda' = \beta_{j-1}, \beta_j$ we have $|\lambda - \lambda'| \leqslant \epsilon$. Also, for these values of $\lambda'$, we have from event $\mathcal{E}_{1,N_t}$:

$$|Z_t^{\lambda'} - \mu^{\mathrm{G}}| \leqslant \sqrt{\frac{2\hat{\mathrm{V}}_t^{\lambda'} L_t}{N_t^{\mathrm{G}}}} + \frac{3(1+\gamma)bL_t}{N_t^{\mathrm{G}}} \tag{7}$$

From (6) and using $\sqrt{x+y} \leqslant \sqrt{x} + \sqrt{y}$ for $x, y > 0$, we also have:

$$\sqrt{\hat{V}_t^{\lambda'}} \leqslant \sqrt{\hat{V}_t^\lambda + \frac{\gamma'(\gamma+4)b^2}{N_t}} \leqslant \sqrt{\hat{V}_t^\lambda} + \sqrt{\frac{\gamma'(\gamma+4)b^2}{N_t}} \leqslant \sqrt{\hat{V}_t^\lambda} + \frac{(\gamma+2)b}{\sqrt{N_t^{\mathrm{G}}}}$$

Using the above bound in (7), we obtain the following for $\lambda' = \beta_{j-1}, \beta_j$

$$|Z_t^{\lambda'} - \mu^{\mathrm{G}}| \leqslant \sqrt{\frac{2\hat{V}_t^\lambda L_t}{N_t^{\mathrm{G}}}} + \frac{(\gamma+2)b\sqrt{2L_t}}{N_t^{\mathrm{G}}} + \frac{3(1+\gamma)bL_t}{N_t^{\mathrm{G}}}$$

$$\leqslant \sqrt{\frac{2\hat{V}_t^\lambda L_t}{N_t^{\mathrm{G}}}} + \frac{6(1+\gamma)bL_t}{N_t^{\mathrm{G}}}, \tag{8}$$

where the last inequality above uses the fact that $L_t \geqslant 1$, which means $\sqrt{L_t} \leqslant L_t$. Now we show the concentration of $Z_t^\lambda$ by noting that $|Z_t^\lambda - \mu^{\mathrm{G}}|$ is a convex function of $\lambda$ and since $\lambda \in I_j$:

$$|Z_t^\lambda - \mu^{\mathrm{G}}| \leqslant \max_{\lambda' \in \{\beta_{j-1}, \beta_j\}} |Z_t^{\lambda'} - \mu^{\mathrm{G}}| \leqslant \sqrt{\frac{2\hat{V}_t^\lambda L_t}{N_t^{\mathrm{G}}}} + \frac{6(1+\gamma)bL_t}{N_t^{\mathrm{G}}}, \tag{9}$$

where we obtain the final bound from (8). Since $\lambda \in [-\gamma', \gamma']$ was arbitrary, (9) holds for all $\lambda \in [-\gamma', \gamma']$. In particular, since the random variable $\lambda_t \in [-\gamma', \gamma']$, we have that:

$$|Z_t^{\lambda_t} - \mu^{\mathrm{G}}| = |Z_t - \mu^{\mathrm{G}}| \leqslant \sqrt{\frac{2\hat{V}_t^{\lambda_t} L_t}{N_t^{\mathrm{G}}}} + \frac{6(1+\gamma)bL_t}{N_t^{\mathrm{G}}} \tag{10}$$

Since $\lambda_t = \arg\min_{\lambda \in [-\gamma', \gamma']} \hat{V}_t^\lambda$ and $\lambda_t^* \in [-\gamma', \gamma']$, we have $\hat{V}_t^{\lambda_t} \leqslant \hat{V}_t^{\lambda_t^*}$. Combining this with the bound in event $\mathcal{E}_{2,N_t}$, we obtain:

$$\sqrt{\hat{V}_t^{\lambda_t}} \leqslant \sqrt{\hat{V}_t^{\lambda_t^*}} \leqslant \sqrt{V_t^{\lambda_t^*}} + b(1+\gamma)\sqrt{\frac{L_t}{2N_t^{\mathrm{G}}}}.$$

Plugging the above into (10) and noting $V_t^{\lambda_t^*} = V_t^*$ gives us the result in the lemma for a fixed number of pulls $N_t(i)$.

To show the anytime version, note that $\mathbb{P}\left(\bigcap_{n \geqslant 1} \mathcal{E}_n\right) \geqslant 1 - \sum_{n \geqslant 1} \mathbb{P}(\mathcal{E}_n^c) \geqslant 1 - \sum_{n \geqslant 1} \delta_n \geqslant 1 - \delta$. Therefore, the anytime concentration result holds w.p. at least $1 - \delta$. This completes the proof. $\quad\square$

### B.4 Proofs for regret of CUUCB

In this subsection we present the proof for Theorem 5.3, which bounds the regret of CUUCB.

*Proof of Theorem 5.3.* We decompose the pseudo-regret as:

$$\mathrm{Regret}(T) = \sum_{i \in [K]: \Delta_i > 0} N_T(i)\Delta_i,$$

where $\Delta_i = \max_{j \in [K]} \mu^{\mathrm{G}}(j) - \mu^{\mathrm{G}}(i)$ and $N_T(i)$ is the total number of times arm $i$ is pulled. We denote $\mu^{*,\mathrm{G}} := \max_{j \in [K]} \mu^{\mathrm{G}}(j)$. Let $i^*$ be an optimal arm, i.e., an arm satisfying $\mu^{\mathrm{G}}(i^*) = \mu^{*,\mathrm{G}}$.

Let us fix a sub-optimal arm $i$. We denote by $\mathcal{E}_i$ the event

$$\mathcal{E}_i = \left\{ |Z_t(j) - \mu^{\mathrm{G}}(j)| \leqslant \sqrt{\frac{2\hat{V}_t^{\lambda_t(j)}(j)L_t(j)}{N_t^{\mathrm{G}}(j)}} + \frac{6(1+\gamma)bL_t(j)}{N_t^{\mathrm{G}}(j)} \leqslant D_t(j), \text{ for } j = i, i^* \right\}$$

where

$$D_t(j) := \sqrt{\frac{2V_t^*(j)L_t(j)}{N_t^{\mathrm{G}}(j)}} + \frac{7(1+\gamma)bL_t(j)}{N_t^{\mathrm{G}}(j)} \tag{11}$$

From Lemma 5.2, the event $\mathcal{E}_i$ occurs with at least $1 - 2\delta$ probability. On this event, we have that $\mu^{\mathrm{G}}(i^*) \leqslant U_t(i^*)$ for all times $t$. Thus, the sub-optimal arm $i$ is not pulled at any time $t$ when $U_t(i) < \mu^{\mathrm{G}}(i^*) \leqslant U_t(i^*)$. We will use this upper bound the total number of pulls $N_T(i)$.

On event $\mathcal{E}_i$, we have:

$$U_t(i) \leqslant Z_t(i) + D_t(i) \leqslant \mu^{\mathrm{G}}(i) + 2D_t(i)$$

Thus arm $i$ is not pulled at any time $t$ when $\mu^{\mathrm{G}}(i) + 2D_t(i) < \mu^{*,\mathrm{G}} \iff D_t(i) < \Delta_i/2$. Now, we upper bound $D_t(i)$.

From here on, we drop the index $i$, which denotes the sub-optimal arm under consideration. From Lemma 4.1, we have $1 - \alpha_t \geqslant (1 - \alpha)\left(1 - \frac{1}{N_t}\right)$. Noting $\mathrm{V}_t^* = \mathrm{Var}^{\mathrm{G}} - (1 - \alpha_t)\frac{\mathrm{Cov}^2}{\mathrm{Var}^{\mathrm{C}}}$, we get

$$\mathrm{V}_t^* \leqslant \mathrm{Var}^{\mathrm{G}} - (1 - \alpha)\frac{\mathrm{Cov}^2}{\mathrm{Var}^{\mathrm{C}}} + (1 - \alpha)\frac{\mathrm{Cov}^2}{\mathrm{Var}^{\mathrm{C}} N_t} \leqslant \mathrm{V}^* + \frac{\gamma^2 b^2}{4N_t},$$

where the last step above uses $|\mathrm{Cov}| \leqslant \gamma \mathrm{Var}^{\mathrm{C}}$ and $\mathrm{Var}^{\mathrm{C}} \leqslant b^2/4$. Taking square root on both sides in the above inequality, we get:

$$\sqrt{\mathrm{V}_t^*} \leqslant \sqrt{\mathrm{V}^* + \frac{\gamma^2 b^2}{4N_t}} \leqslant \sqrt{\mathrm{V}^*} + \frac{\gamma b}{2\sqrt{N_t}} \leqslant \sqrt{\mathrm{V}^*} + \frac{\gamma b}{2\sqrt{N_t^{\mathrm{G}}}} \tag{12}$$

Substituting (12) into (11), we obtain:

$$D_t \leqslant \sqrt{\frac{2\mathrm{V}^* L_t}{N_t^{\mathrm{G}}}} + \frac{\gamma b\sqrt{L_t/2}}{N_t^{\mathrm{G}}} + \frac{7(1 + \gamma)bL_t}{N_t^{\mathrm{G}}}$$

$$< \sqrt{\frac{2\mathrm{V}^* L_t}{N_t^{\mathrm{G}}}} + \frac{8(1 + \gamma)bL_t}{N_t^{\mathrm{G}}}$$

where in the last step, we use $L_t \geqslant 1$.

Thus, a sufficient condition for $D_t < \Delta/2$ is:

$$\sqrt{\frac{2\mathrm{V}^* L_t}{N_t^{\mathrm{G}}}} + \frac{8(1 + \gamma)bL_t}{N_t^{\mathrm{G}}} \leqslant \frac{\Delta}{2}$$

$$\Leftarrow \frac{2\mathrm{V}^* L_t}{N_t^{\mathrm{G}}} \leqslant \frac{\Delta^2}{16} \text{ and } \frac{8(1 + \gamma)bL_t}{N_t^{\mathrm{G}}} \leqslant \frac{\Delta}{4}$$

$$\iff \frac{N_t^{\mathrm{G}}}{L_t} \geqslant \max\left\{\frac{32\mathrm{V}^*}{\Delta^2}, \frac{32(1 + \gamma)b}{\Delta}\right\}$$

$$\Leftarrow N_t^{\mathrm{G}} \geqslant \frac{32}{\Delta}\left(\frac{\mathrm{V}^*}{\Delta} + (1 + \gamma)b\right)L_t$$

$$\overset{(i)}{\Leftarrow} \alpha(N_t - 1) \geqslant \frac{128}{\Delta}\left(\frac{\mathrm{V}^*}{\Delta} + (1 + \gamma)b\right)\log(2T)$$

Implication $(i)$ uses Lemma 4.1 which gives us $N_t^{\mathrm{G}} = \alpha_t N_t \geqslant \alpha(1 - N_t^{-1})N_t = \alpha(N_t - 1)$. It also uses the bound $L_t \leqslant \log(16\delta^{-1}(N_t + 1)^3) \leqslant 4\log(2T)$ for any $t \in [T]$. Thus if

$$N_t \geqslant \frac{128}{\alpha\Delta}\left(\frac{\mathrm{V}^*}{\Delta} + (1 + \gamma)b\right)\log(2T) + 1,$$

then arm $i$ is not pulled. Reintroducing the index $i$, this means we obtain the following bound on the number of pulls of arm $i$ when $\mathcal{E}_i$ occurs:

$$N_T(i) \leqslant \frac{128}{\alpha(i)\Delta_i}\left(\frac{\mathrm{V}^*(i)}{\Delta_i} + (1 + \gamma)b\right)\log(2T) + 2 \tag{13}$$

Thus, we have

$$\mathbb{E}[N_T(i)] = \mathbb{E}[N_T(i)\mathbb{1}\{\mathcal{E}_i\}] + \mathbb{E}[N_T(i)\mathbb{1}\{\mathcal{E}_i^c\}]$$

$$\leqslant \frac{128}{\alpha(i)\Delta_i}\left(\frac{\mathrm{V}^*(i)}{\Delta_i} + (1+\gamma)b\right)\log(2T) + 2 + T\mathbb{P}(\mathcal{E}_i^c)$$

We obtain this inequality by using (13) on event $\mathcal{E}_i$ along with the trivial bound $N_T(i) \leqslant T$ on event $\mathcal{E}_i^c$. Using $\mathbb{P}(\mathcal{E}_i^c) \leqslant 2/T$ gives us:

$$\mathbb{E}[N_T(i)] \leqslant \frac{128}{\alpha(i)\Delta_i}\left(\frac{\mathrm{V}^*(i)}{\Delta_i} + (1+\gamma)b\right)\log(2T) + 4 \tag{14}$$

Using (14), we can bound the expected regret of the algorithm as:

$$\mathbb{E}[\mathrm{Regret}(T)] = \sum_{i\in[K]} \Delta_i \mathbb{E}[N_T(i)]$$

$$\leqslant \sum_{i:\Delta_i>0}\left(\frac{128}{\alpha(i)}\left(\frac{\mathrm{V}^*(i)}{\Delta_i} + (1+\gamma)b\right)\log(2T) + 4\Delta_i\right)$$

This gives us the instance-dependent regret bound. For the instance-independent regret bound, consider some $\hat{\Delta} > 0$. While bounding the expected, we bound the constributions of arms $i$ with $\Delta_i > \hat{\Delta}$ and arms $i$ with $\Delta_i \leqslant \hat{\Delta}$ differently as follows (Note $\Delta_i \leqslant b$ for any $i$):

$$\mathbb{E}[\mathrm{Regret}(T)] \leqslant \sum_{i:\Delta_i>\hat{\Delta}}\left(\frac{128}{\alpha(i)}\left(\frac{\mathrm{V}^*(i)}{\hat{\Delta}} + (1+\gamma)b\right)\log(2T) + 4b\right) + \sum_{i:\Delta_i\leqslant\hat{\Delta}}\hat{\Delta}\mathbb{E}[N_T(i)]$$

$$\leqslant \frac{128\log(2T)}{\hat{\Delta}}\left(\sum_i \frac{\mathrm{V}^*(i)}{\alpha(i)}\right) + \hat{\Delta}T + 132\sum_i \alpha(i)^{-1}(1+\gamma)b\log(2T)$$

Choosing $\hat{\Delta} = \sqrt{128\left(\sum_i \frac{\mathrm{V}^*(i)}{\alpha(i)}\right)\frac{\log(2T)}{T}}$ to minimize the upper bound gives us:

$$\mathbb{E}[\mathrm{Regret}(T)] \leqslant 2\sqrt{128\left(\sum_i \frac{\mathrm{V}^*(i)}{\alpha(i)}\right)T\log(2T)} + \mathcal{O}(\log(T))$$

$$\leqslant 23\sqrt{\left(\sum_i \frac{\mathrm{V}^*(i)}{\alpha(i)}\right)T\log(2T)} + \mathcal{O}(\log(T))$$

This completes the proof. $\qquad\square$

**Lemma B.3** (Regret bound for UCB-V). *When* UCB-V *is implemented with the threshold rule* (2), *while using only ground-truth samples for decision making, then its expected regret satisfies the following instance-independent bound:*

$$\mathbb{E}[Regret(T)] \leqslant \mathcal{O}\left(\sqrt{\left(\sum_i \frac{\mathrm{Var}^{\mathrm{G}}(i)}{\alpha(i)}\right)T\log T}\right)$$

*Proof of Lemma B.3.* When UCB-V is run using only ground-truth samples, then the estimated upper-confidence bound for arm $i$ at time $t$ by UCB-V(Audibert et al., 2009) is given by (barring constants, which do not affect the asymptotic order of regret):

$$U_t'(i) = \mu_t^{\mathrm{G}}(i) + \sqrt{\frac{2\mathrm{Var}_t^{\mathrm{G}}(i)L_t'(i)}{N_t^{\mathrm{G}}(i)}} + \frac{3bL_t'(i)}{N_t^{\mathrm{G}}(i)},$$

where $L_t'(i)$ is an exploration function typically taken to be growing as $\log t$. We fix the arm $i$ drop the index $i$ in the following analysis. Let us take $\delta_n = \frac{\delta}{2n^2}$ We see by taking $\lambda = 0, \gamma = 0$ in Lemma

B.2 and Lemma B.1, and using union bound that the following concentration inequality holds after $N_t^{\mathrm{G}}$ pulls:

$$|Z_t^0 - \mu^{\mathrm{G}}| \leqslant \sqrt{\frac{2\hat{V}_t^0 \log\left(10/\delta_{N_t^{\mathrm{G}}}\right)}{N_t^{\mathrm{G}}}} + \frac{3b \log\left(10/\delta_{N_t^{\mathrm{G}}}\right)}{N_t^{\mathrm{G}}} \leqslant \sqrt{\frac{2V_t^0 \log\left(10/\delta_{N_t^{\mathrm{G}}}\right)}{N_t^{\mathrm{G}}}} + \frac{4b \log\left(10/\delta_{N_t^{\mathrm{G}}}\right)}{N_t^{\mathrm{G}}},$$

w.p. at least $1 - \delta_{N_t^{\mathrm{G}}}/5 - 4\delta_{N_t^{\mathrm{G}}}/5 = 1 - \delta_{N_t^{\mathrm{G}}}$. We note that $Z_t^0 = \mu_t^{\mathrm{G}}$, $\hat{V}_t^0 = \mathrm{Var}_t^{\mathrm{G}}$ and $V_t^0 = \mathrm{Var}^{\mathrm{G}}$. Since $\sum_n \delta_n \leqslant \delta$, we have that the above inequality holds for all times w.p. at least $1 - \delta$, which gives us the following anytime concentration:

$$|\mu_t^{\mathrm{G}} - \mu^{\mathrm{G}}| \leqslant \sqrt{\frac{2\mathrm{Var}_t^{\mathrm{G}} L_t'}{N_t^{\mathrm{G}}}} + \frac{3b L_t'}{N_t^{\mathrm{G}}} \leqslant \sqrt{\frac{2\mathrm{Var}^{\mathrm{G}} L_t'}{N_t^{\mathrm{G}}}} + \frac{4b L_t'}{N_t^{\mathrm{G}}} \tag{15}$$

Where we have set $L_t' = \log(20(N_t^{\mathrm{G}})^2/\delta)$, and we will show the regret bound setting $\delta = 1/T$. This choice of $L_t'$ matches the $\log t$ growth we want. On this event, we have that

$$U_t(i) \leqslant \mu^{\mathrm{G}}(i) + 2D_t'(i)$$

for $D_t'(i) = \sqrt{\frac{2\mathrm{Var}^{\mathrm{G}}(i)L_t'(i)}{N_t^{\mathrm{G}}(i)}} + \frac{4b L_t'(i)}{N_t^{\mathrm{G}}(i)}$.

Now, we proceed identically as in the regret proof of Lemma 5.3, by considering the event $\mathcal{E}_i$ w.p. at least $1 - 2/T$, when the anytime-concentration (15) holds for the sub-optimal arm $i$ and some optimal arm $i^*$. On event $\mathcal{E}_i$, we bound the number of pulls after which $U_t(i) < \mu^{\mathrm{G}}(i^*)$, which happens if $D_t'(i) < \Delta_i/2$. This is achieved at times $t$ such that:

$$N_t^{\mathrm{G}}(i) \geqslant \frac{32 L_t'(i)}{\Delta_i}\left(\frac{\mathrm{Var}^{\mathrm{G}}(i)}{\Delta_i} + b\right)$$

$$\Leftarrow \alpha(i)(N_t(i) - 1) \geqslant \frac{C \log T}{\Delta_i}\left(\frac{\mathrm{Var}^{\mathrm{G}}(i)}{\Delta_i} + b\right),$$

which follows from Lemma 4.1 and bounding $L_t'(i) \leqslant C \log T$, for an absolute constant $C$ and $T$ sufficiently large. Thus, proceeding as in the derivation of (14),

$$\mathbb{E}[N_T(i)] \leqslant \frac{C}{\alpha(i)\Delta_i}\left(\frac{\mathrm{Var}^{\mathrm{G}}(i)}{\Delta_i} + b\right)\log(T) + 4 \tag{16}$$

Using the same argument as in the derivation of the instance-independent regret bound in the proof of Lemma 5.3, we obtain the following instance-independent bound for UCB-V

$$\mathbb{E}[\mathrm{Regret}(T)] \leqslant 2\sqrt{C\left(\sum_i \frac{\mathrm{Var}^{\mathrm{G}}(i)}{\alpha(i)}\right)T\log(T)} + \mathcal{O}(\log(T))$$

which concludes the proof. $\qquad\square$

## C EXPERIMENTAL DETAILS

### C.1 IMPLEMENTATION OF ALGORITHMS

To make the comparison with other UCB variants more straightforward, we implement the upper confidence bound $U_t(i)$ for CUUCB a little differently than described in the theoretical analysis while preserving its main structure. The implemented expressions of $U_t(i)$ for each arm $i$ at time $t$ for each algorithm is given below:

$$\mathrm{UCB} : U_t(i) = \mu_t^{\mathrm{G}}(i) + \sqrt{\frac{\zeta_1 b^2 \log t}{N_t^{\mathrm{G}}(i)}}$$

$$\text{UCB-V} : U_t(i) = \mu_t^{\text{G}}(i) + \sqrt{\frac{2\text{Var}_t^{\text{G}}(i)\log t}{N_t^{\text{G}}(i)}} + \frac{b\zeta_2 \log t}{N_t^{\text{G}}(i)}$$

$$\text{CUUCB} : U_t(i) = Z_t(i) + \sqrt{\frac{2\hat{\text{V}}_t^{\lambda_t(i)}(i)\log t}{N_t^{\text{G}}(i)}} + \frac{b\zeta_3 \log t}{N_t^{\text{G}}(i)}$$

Here $\{\zeta_j\}_{j=1}^3$ are tunable parameters. Moreover, we obtain the final upper confidence bound by taking $\min\{U_t(i), b\}$ in each case. Note that UCB and UCB-V use only ground-truth observations in their confidence bounds. Thus, all algorithms use rule (2) to pick the set of ground-truth measurements and choose arms for all $T$ time steps. Also note that the implementation of UCB-V is different from the one described above in the LLM-as-a-judge experiment on the mis-aligned Dataset-B, where we run UCB-V using only coarse feedback at all time steps. So, in just this experiment, the expression for $U_t(i)$ for UCB-V is given by:

$$U_t(i) = \hat{\mu}_t^{\text{C}}(i) + \sqrt{\frac{2\hat{\text{V}}_t^{\text{C}}(i)\log t}{N_t(i)}} + \frac{b\zeta_2 \log t}{N_t(i)},$$

where

$$\hat{\mu}_t^{\text{C}}(i) = \frac{1}{N_t(i)} \sum_{j \in [t]:i_j=i} r_j^{\text{C}}(i), \qquad \hat{\text{V}}_t^{\text{C}}(i) = \frac{1}{N_t(i)} \sum_{j \in [t]:i_j=i} (r_j^{\text{C}}(i) - \hat{\mu}_t^{\text{C}}(i))^2$$

are empirical mean and variance estimators of all the coarse measurements observed for arm $i$ at time $t$.

In all of our experiments, we used $\zeta_1 = 1, \zeta_2 = 0.5, \zeta_3 = 0.5$. We also use a uniform target ratio $\alpha(i) = \alpha$ for all arms $i$ in all our experiments.

## C.2 DATA GENERATION

We describe the dataset generation methods used for our experiments. Our generated datasets are always a collection of $N$ 2-D vectors for each of the $K$ arms. We refer to $N$ as the size of the dataset. When running algorithms on a generated dataset, the reward pair $[r_t^{\text{G}}(i), r_t^{\text{C}}(i)]$ for an arm $i$ is generated by sampling uniformly from this dataset. Thus, the mean rewards $[\mu^{\text{G}}(i), \mu^{\text{C}}(i)]$ are taken to be the empirical means of the dataset.

### C.2.1 SYNTHETIC

For the synthetic experiments, our dataset generation is parameterized by the correlation parameter $\rho$. The generation of a (ground-truth, coarse) measurement pair for a give arm $i$ proceeds as follows:

1. Sample from a 2-D Gaussian with mean $[\mu_1(i), \mu_2(i)]$ and covariance matrix $C(\rho)$.

2. Clip the values in the interval $I$.

We set $C(\rho) = \begin{bmatrix} 5 & 4.5\rho \\ 4.5\rho & 4 \end{bmatrix}$.

We fix the means $\mu_1(i)$ for $i = 1, ..., 4$ as $[0.5, 0.7, 0.8, 0.2]$. We then obtain $\mu_2(i)$ as a perturbed version of $\mu_1(i)$ by uniformly sampling in the interval $\mu_1(i) \pm 0.1$ and then clipping in the interval $[0, 1]$. We also set $I = [-4, 6]$.

For each value of $\rho$, we generate the dataset by sampling $N = 100,000$ datapoints as described above for each arm $i$.

## C.3 LLM-AS-A-JUDGE

We make use of the Nectar dataset of rankings of (prompt,answer) pairs from Zhu et al. (2024) for our dataset generation. This dataset comprises of 182,954 prompts, each answered by seven LLMs. These answers are then scored by GPT-4 Bubeck et al. (2023).

To generate our dataset, we extract a subset of the data having (prompt, answer) pairs for six chosen models and use a pair of LLMs to score the models. Crucially, we make use of the same scoring rubric and prompt while querying both LLMs. Details of the chosen models, exact prompt and rubrics used can be found in the code attached in the supplementary material. We treat the scores given by the more advanced LLM as the ground-truth (i.e., proxies for human labels which are costly to collect at scale) and those by the less advanced LLM as coarse feedback.

By this method, we obtain Dataset-A of size 19,089 by querying LLama 3 3b (strong model) and LLama 3 1B (weak model) for each of the six models. We similarly obtain Dataset-B of size 5000 using scores from Gemini 2.5-Flash (strong model) and Gemini 1.5-Flash (weak model). The datasets can be found in the attached code.

## C.4 ADDITIONAL EXPERIMENTAL RESULTS

### C.4.1 SYNTHETIC

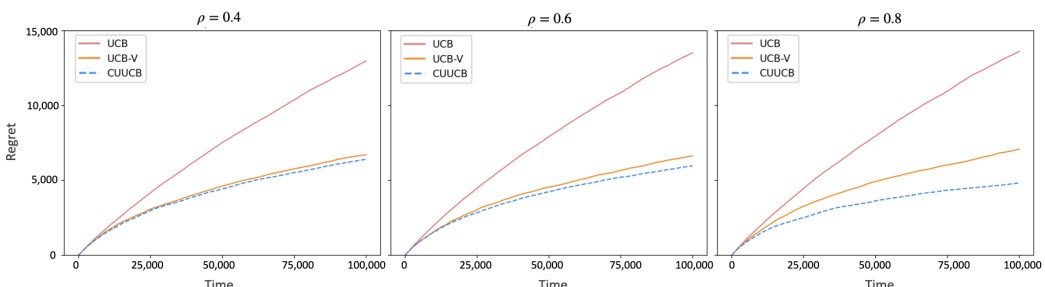

Figure 3: Comparison of CUUCB, UCB and UCB-V on synthetically generated data for a fixed value of $\alpha = 0.1$ and different values of $\rho$ with a time horizon of $100,000$ steps, averaged across $40$ runs.

We also run the synthetic experiments with fixed $\alpha = 0.1$ and varying $\rho$. We see from Figure 3 that increasing $\rho$ leads to an improvement in the performance of CUUCB, which matches our expectations from theory. Since UCB and UCB-V only use ground-truth data, changing $\rho$ does not affect their performance.

### C.4.2 LLM-AS-A-JUDGE

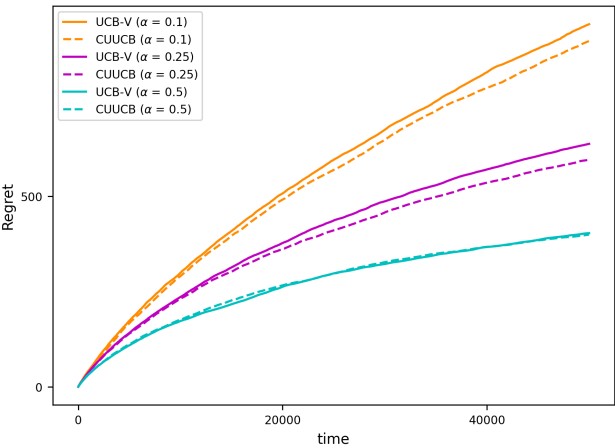

Figure 4: LLM benchmarking regret plots on Dataset-A of UCB-V which has access to only ground truth samples vs. CUUCB for different $\alpha$, averaged across $40$ runs.

We also run an experiment where we compare regrets of CUUCB and UCB-V on Dataset-A while varying the parameter $\alpha$ as shown in Figure 4. We see that our algorithm CUUCB achieves only a small improvement over UCB-V even for small $\alpha$, while the improvement was more significant when

Dataset-B was used, as seen from Figure 2b. This shows that the performance gains of CUUCB are sensitive to the dataset instance.

