# OpenReview forum: "Learning with Coupled Uncertainty"
_ICLR.cc/2026/Conference — Submitted to ICLR 2026_

### Official Review · Reviewer_5GF4 · 2025-10-25

**Soundness:** 2
**Presentation:** 1
**Contribution:** 2
**Rating:** 2
**Confidence:** 3

**Summary:**

The paper studies a multi-fidelity K-armed bandit problem where, at each pull, the learner may obtain a cheap coarse measurement and, on a controlled fraction of pulls, an expensive ground-truth measurement. When ground truth is queried, the corresponding coarse measurement is also observed. The authors propose CUUCB, a UCB-style algorithm that uses a variance-reduced estimator to combine the two feedback types, while enforcing a per-arm cap on the fraction of ground truth queries. They prove instance-dependent and instance-independent regret bounds whose leading constant improves with the correlation between coarse and ground truth measurements. Experiments on synthetic data and an LLM-as-a-judge benchmarking task show regret improvements over UCB-V under limited ground truth budgets and when there is misalignment.

**Strengths:**

1. The paper provides a clear link to practical applications like LLM evaluation.

2. Variance-reduced estimators are clearly defined, and regret analysis looks rigorous. The analysis shows improvement over baseline UCB-V

3. Given the importance of prediction-power inference, this paper represents a timely problem that is of interest to the greater ICLR community.

**Weaknesses:**

1. Reading the first three sections first gives the reader (wrong) the sense that this paper is addressing a very general problem in the contextual bandit setting. The motivating examples are indeed contextual in nature. Section 4 then suddenly jumps to the K-armed stochastic bandit, which is not aligned with the expectations that the first three sections build. The motivation is strong and broad, but the contribution of the paper does not correctly align with its motivation. The scope of the solution is narrow. It is also not clear how the findings (estimators and algorithm) can be extended to contextual setups and setups with a large number of arms.

2. A known upper bound on the variance ratio is a strong assumption. Discussion of robustness to misspecification would strengthen the work.

3. The regret bound improves over UCB-V by exploiting coupled uncertainties in the upper confidence bound, but there is no accompanying lower bound. Without a lower bound, it is not clear if what is done optimally utilizes the correlations between ground truth and coarse measurements.

4. The simple thresholding rule decouples when to sample ground-truth from when to choose a particular arm. While it simplifies the algorithm design, it is not clear what is lost in terms of optimality with such a choice. It also aims for an anytime guarantee on the ratio of ground truths obtained. Since the time horizon is known (input to Algorithm 1), why should one enforce an anytime guarantee on the ground truth ratio? For instance, in the LLM example, having a ground truth budget is more realistic (and less stringent) than requiring an anytime guarantee on the ground truth ratio.

5. Experiments are somewhat limited in scope. In particular, LLMs are used as ground truth proxies rather than human labels.

6. Experiments consider a very small number of arms. How will the improvement scale as the number of arms increase?

**Questions:**

1. How sensitive is CUUCB to misspecification of the $\gamma$ parameter?

2. Can the framework extend naturally to contextual bandits or reinforcement learning settings? The introduction hints at this but does not elaborate.

3. How robust is CUUCB when coarse and ground-truth data are weakly correlated or adversarially misaligned?

---

> ### Author Response · Authors · 2025-11-24
> **Robustness to misspecification of $\gamma$ parameter**
>
> We can, in fact, derive regret bounds for CUUCB without assumption 4.2, which allows us to see the loss in performance if the variance ratio $\gamma$ is mis-specified. Essentially, the assumption we really use in our proof is that $\frac{|\text{Cov}(i)|}{\text{Var}^\text{C}(i)} \le \gamma$. If we do not make this assumption, we only need to modify one step in the proof and obtain slightly modified regret bounds as described below.
>
> Define $\kappa(i):= \frac{|\text{Cov}(i)|}{\text{Var}^\text{C}(i)}$, and let
> $$\text{V}^\gamma(i):=
> \begin{cases}
> \text{Var}^\text{G}(i) - (1-\alpha)\gamma (2\kappa(i)- \gamma)\text{Var}^\text{C}(i), \text{ if }\gamma < \kappa(i) \\\\
> \text{V}^\*(i), \text{ otherwise}
> \end{cases}$$
> Then the regret bounds for CUUCB have the form:
>
> Instance-Dependent Regret Bound:
>      \begin{equation*}
>         \mathbb{E}[\text{Regret}(T)]
>         \le 128\sum_{i:\Delta_i>0} \frac{1}{\alpha(i)} \left(\frac{\text{V}^{\gamma}(i)}{\Delta_i} + b(1+\gamma+ \kappa(i)) \right)\log(2T)
>      + \mathcal{O}(1),
>     \end{equation*}
>  Instance-Independent Regret Bound:
>      $$ \mathbb{E}[\text{Regret}(T)]
>         \le  23\sqrt{\left(\sum_{i} \frac{\text{V}^\gamma(i)}{\alpha(i)}\right)  T \log(2T)} + \mathcal{O}(\log(T))$$
> We can include a formal proof of the above results in a revised version as needed. We provide some intuition on the quantity $\text{V}^\gamma$ in the case when $\gamma < \kappa(i)$ for some arm $i$. For the remainder of the discussion, we drop the arm index $i$ for brevity.
>
> Intuitively, the above result is a consequence of the estimator $\lambda_t$ being truncated in the interval $[-\gamma', \gamma']$ (where $\gamma':=\gamma (1-\alpha_t)$). When $\gamma<\kappa$, this interval does not contain the optimal value $\lambda_t^\*$. Thus, the resulting minimum variance factor that is obtained in the concentration radius is given by
>
> $$\min_{\lambda \in [-\gamma', \gamma']} \text{V}_t^\lambda = \text{V}_t^{\gamma' \text{sgn(Cov)}} = \text{Var}^\text{G} - (1-\alpha_t)\gamma (2\kappa- \gamma)\text{Var}^\text{C},$$
>
> which has a limiting value $\text{V}^\gamma$ as the number of samples grows large. Thus, $\text{V}^\gamma$ replaces the optimal variance $\text{V}^\*$ in the regret bounds.
>
> Also note that as $\gamma$ approaches $\kappa$, we see that $\text{V}^\gamma$ approaches $\text{V}^\*$. Clearly, this bound demonstrates robustness to misspecification of $\gamma$, since we have:
> $$\text{V}^\gamma - \text{V}^\* = (1-\alpha)(\kappa-\gamma)^2\text{Var}^\text{C}.$$
> So if the misspecification of $\gamma$ with respect to the true value $\kappa$ is small, there is only a small increase in the variance. Also note that $\text{V}^\gamma \le \text{Var}^\text{G}$, which means we still obtain a variance reduction over using no coarse samples.
>
> In short, the misspecification of $\gamma$ causes CUUCB to use a clipped estimate of $\lambda^*_t$, which affects the resulting variance reduction in the manner described above.

---

> ### Author Response · Authors · 2025-11-24
> **General comment**
>
> We want to thank you for your valuable feedback. We would like to clarify and reiterate some of our contributions
> 1. We pose a new problem with this form of feedback and highlight its potential uses in online learning. This is a general conceptual contribution that can clearly be built upon in contextual bandits, RL, etc.
> 2. We perform a first theoretical treatment of this feedback structure, see how we can leverage the coupling to achieve better performance, and give some theoretical backing to the framework.
> 3. We provide numerical experiments showing the promise of this in synthetic experiments---albeit ones that use real data.
>
> We would also like to address technical contributions in our analysis. Since the covariance structure is unknown, we construct an empirical estimator $\lambda_t$ which is used in the AIPW estimator. Although the AIPW is a well-studied estimator, $\lambda_t$ being a random variable means that the concentration of the resulting AIPW estimator does not straightforwardly follow from Bennett’s concentration. It requires a careful covering argument along with a truncation built into the definition of $\lambda_t$, which ensures boundedness.

---

> ### Author Response · Authors · 2025-11-25
> **Justification for the choice of an anytime guarantee on ground truth ratio**
>
> (In response to weakness 4)  There are standard techniques one can use to modify CUUCB to obtain an anytime algorithm. These techniques use martingale-type concentration bounds and can be implemented without a significant change to the main proof idea for CUUCB. As an example of such techniques, see the proof ideas for the UCB-V algorithm from the paper "Exploration–exploitation tradeoff using variance estimates in multi-armed bandits" (Audibert et al., 2009).

---

> ### Author Response · Authors · 2025-11-25
> **Using "advanced LLM" as proxy for human labels**
>
> It is not necessary for the “advanced” LLM responses to precisely match human responses to carry out our experiment – the main idea is to show that CUUCB can debias the coarse feedback using a small amount of ground feedback, which is chosen to be the “advanced” LLM response here. The experiments serve as a proof of concept, since we do not have access to real human responses to the prompts. The ‘more advanced LLM’ is to be thought of as a model for human responses. The misalignment between the ‘advanced’ and ‘simple’ LLM response models the misalignment one might find between human and LLM responses. For some motivation on using LLMs to model human feedback, see the paper “Correlating and Predicting Human Evaluations of Language Models from Natural Language Processing Benchmarks” (Schaeffer et al., 2025).

---

> > ### Comment · Reviewer_5GF4 · 2025-11-27
> >
> > Thanks for the response. Extending the analysis to handle misspecified gamma is an important contribution. The paper will benefit from it if regret bounds with rigorous proofs can be given for this case. Working with LLMs as human proxies makes the experiments easy to conduct. On the other hand, there are still major concerns related to the optimality of the regret bounds (no lower bound), and it is unclear whether the simple thresholding rule that decouples when to sample ground-truth from when to choose a particular arm is the optimal way. Related to the anytime guarantee, I was asking for clarification on why the anytime feedback ratio guarantee (e.g., Lemma 4.1) is enforced. For instance, in the LLM example, having a ground truth budget is more realistic (and less stringent) than requiring an anytime guarantee on the ground truth ratio.

---

> > > ### Author Response · Authors · 2025-12-01
> > >
> > > Thank you for your response.
> > > We will add the formal proofs for the regret bounds in the case of misspecified $\gamma$.
> > >
> > > As for lower bounds, there are standard proof techniques one can use to obtain variance-dependent lower bounds for Gaussian distributions (which we can approximate with bounded random variables), yielding a suitable worst-case lower bound. For a reference to the proof techniques, see section 15.2 (Minimax Lower Bounds) in the book "Bandit Algorithms" by Tor Lattimore and Csaba Szepesvári.
> > >
> > > As for the anytime feedback ratio guarantee, we enforce it because the algorithm may be used in a setting where the horizon $T$ is not known beforehand, and we just fix the fraction of ground samples as our budget. This is why we mention techniques one can employ to make CUUCB an anytime algorithm. It is true that in some scenarios, we might want to look at other ways of budgeting the ground samples, which will require a separate analysis to come up with an appropriate decision rule.

---

### Official Review · Reviewer_H752 · 2025-10-26

**Soundness:** 3
**Presentation:** 3
**Contribution:** 3
**Rating:** 6
**Confidence:** 2

**Summary:**

This paper investigates decision-making under coupled uncertainties, where a learner has access to both ground-truth and coarse measurements of outcomes. The authors introduce a model in which these two types of feedback are statistically coupled, and the proposed approach learns their correlation to optimally combine coarse measurements with limited ground-truth feedback for improved performance. The method leverages confidence bounds that exploit the correlation structure between the measurements to reduce uncertainty and enhance decision accuracy.

**Strengths:**

This paper is the first to explore the use of coupled random variables through ground-truth and coarse measurements in a bandit framework. The proposed approach effectively leverages the correlation between these measurements to extract additional information from coarse feedback, resulting in improved performance.

**Weaknesses:**

The main concern is the novelty of the algorithm. The proposed method heavily relies on the AIPW estimator, which has been extensively studied in prior work, including Robins & Rotnitzky (1995) and more recently Angelopoulos et al. (2024).


To strengthen the positioning of the contribution, it would also be beneficial to include relevant bandit literature with offline samples in the related work section.

**Questions:**

Could you elaborate on the novelty of the proposed algorithm and its technical contributions relative to prior work on AIPW-based methods?

---

> ### Author Response · Authors · 2025-11-24
> **General Comment**
>
> We want to thank you for your valuable feedback. We would like to clarify and reiterate some of our contributions
> 1. We pose a new problem with this form of feedback and highlight its potential uses in online learning. This is a general conceptual contribution that can clearly be built upon in contextual bandits, RL, etc.
> 2. We perform a first theoretical treatment of this feedback structure, see how we can leverage the coupling to achieve better performance, and give some theoretical backing to the framework.
> 3. We provide numerical experiments showing the promise of this in synthetic experiments---albeit ones that use real data.
>
> We would also like to address technical contributions in our analysis. Since the covariance structure is unknown, we construct an empirical estimator $\lambda_t$ which is used in the AIPW estimator. Although the AIPW is a well-studied estimator, $\lambda_t$ being a random variable means that the concentration of the resulting AIPW estimator does not straightforwardly follow from Bennett’s concentration. It requires a careful covering argument along with a truncation built into the definition of $\lambda_t$, which ensures boundedness.

---

> > ### Comment · Reviewer_H752 · 2025-11-25
> >
> > Thank you for your responses. Could you provide a more detailed comparison with the line of research that incorporates offline samples together with online data in bandit problems?

---

### Official Review · Reviewer_wzTm · 2025-10-31

**Soundness:** 2
**Presentation:** 3
**Contribution:** 2
**Rating:** 4
**Confidence:** 3

**Summary:**

The paper studies a bandit setting with coupled uncertainty, where coarse and ground-truth feedback are correlated within the same arm. This assumption is relevant to recent RLHF-style problems where proxy and human feedback overlap. The authors extend the UCB framework with an explicit covariance estimator and provide finite-sample regret bounds. The paper is clearly written and theoretically sound, but the contribution is incremental: the formulation is close to existing correlated and multi-fidelity bandits, and the experiments are small in scale without strong baselines.

**Strengths:**

**1. Correlated uncertainty assumption is relevant to recent RLHF research**

The assumption that coarse and ground-truth feedback are correlated is realistic and aligns well with recent work in reinforcement learning from human feedback, where model-generated and human-provided signals often overlap but differ in precision. This framing connects the problem to practical and timely settings such as AI-assisted evaluation or LLM-as-a-judge feedback.

**2. Clear extension of UCB using an explicit covariance estimator**

The paper extends the standard UCB framework by explicitly estimating and incorporating the covariance between coarse and ground-truth feedback. The modification is simple but coherent, illustrating how correlated signals can be used to refine exploration in a principled way. The algorithm is clearly presented and easy to understand.

**3. Complete and well-structured theoretical analysis**

The paper provides finite-sample concentration bounds and regret guarantees under standard assumptions. The analysis is internally consistent and clearly written, reflecting a solid grasp of the theory. While the results build on established techniques, the presentation is thorough and self-contained.

**Weaknesses:**

**1. The distinction from correlated and multi-fidelity bandits is conceptually minor**

The paper introduces coupled uncertainty as a new setting where coarse and ground-truth feedback are correlated within the same arm. While this idea is clearly stated, similar correlation structures have been studied in related bandit frameworks. The difference between intra-arm and cross-arm correlation is understandable, but the paper does not convincingly show that it leads to fundamentally new challenges or insights. The conceptual contribution feels incremental rather than fundamental.

**2. Limited methodological novelty and technical depth**

The approach estimates the covariance between coarse and ground-truth feedback and integrates it into a UCB-style algorithm. This design follows well-known variance-reduction ideas from existing inference and bandit literature. The theoretical results are correct and clearly written but rely on familiar concentration and regret analyses. The work extends established methods in a careful but modest way, without introducing new technical insights or algorithmic innovations.

**3. Empirical validation is shallow and lacks comparison with relevant baselines**

The experiments are limited to a toy synthetic setting and a small-scale LLM-as-a-judge example. These results illustrate the idea but do not convincingly demonstrate practical advantages. Comparisons are restricted to UCB and UCB-V, while more relevant baselines such as correlated Gaussian-process bandits, multi-fidelity UCB variants, or other structured methods are not included. Without such baselines, it is difficult to evaluate whether the proposed algorithm offers meaningful empirical improvements.

**Questions:**

See the weaknesses.

---

> ### Author Response · Authors · 2025-11-24
> **General Comment**
>
> We want to thank you for your valuable feedback. We would like to clarify and reiterate some of our contributions
> 1. We pose a new problem with this form of feedback and highlight its potential uses in online learning. This is a general conceptual contribution that can clearly be built upon in contextual bandits, RL, etc.
> 2. We perform a first theoretical treatment of this feedback structure, see how we can leverage the coupling to achieve better performance, and give some theoretical backing to the framework.
> 3. We provide numerical experiments showing the promise of this in synthetic experiments---albeit ones that use real data.
>
> We would also like to address technical contributions in our analysis. Since the covariance structure is unknown, we construct an empirical estimator $\lambda_t$ which is used in the AIPW estimator. Although the AIPW is a well-studied estimator, $\lambda_t$ being a random variable means that the concentration of the resulting AIPW estimator does not straightforwardly follow from Bennett’s concentration. It requires a careful covering argument along with a truncation built into the definition of $\lambda_t$, which ensures boundedness.

---

### Official Review · Reviewer_5jx2 · 2025-11-01

**Soundness:** 3
**Presentation:** 2
**Contribution:** 2
**Rating:** 2
**Confidence:** 3

**Summary:**

This paper investigates decision-making when “coupled uncertainties” are present, a setting where high-quality (ground truth) and lower-quality (coarse) measurements are available for each decision. The authors introduce a formal framework modeling the correlation between the two uncertainty sources, propose CUUCB, a variance-adaptive upper confidence bound bandit algorithm that empirically estimates this correlation and combines both sources to reduce regret. Theoretical analyses show instance-dependent and instance-independent regret bounds, and experiments on synthetic data and LLM-based benchmarking tasks provide empirical validation.

**Strengths:**

- The authors present a well-motivated formalization of decision-making with coupled uncertainty, capturing practical scenarios where high- and low-fidelity feedback is available (e.g., LLM-as-a-judge, scientific experiments).
- Detailed regret bounds are provided (Theorem 5.3), including explicit dependence on the unknown correlation coefficients.

**Weaknesses:**

1. While the main regret analysis is technically strong, the presentation repeatedly relies on boundedness (Assumption 4.1) and a known, global variance ratio bound (Assumption 4.2).
2. Almost all experiments focus on synthetic Gaussian mixtures or LLM ranking datasets where the coarse and ground-truth are directly constructed. It’s unclear whether the benefits of CUUCB persist for realistic scenarios where feedback is more complex or structured (for example, heterogeneous or contextual bandits).
3. The LLM benchmarking task defines ground-truth using a ‘more advanced LLM’ rather than real human labels, which may bias the coarse/ground-truth correlation estimation.
4. The paper occasionally overstates generalizability, stating that the “framework unifies several settings, like multi-fidelity learning, AI delegation, etc.” without presenting evidence in all such scenarios. There is little discussion of when the approach could underperform (e.g., when coarse and ground-truth are misaligned and weakly correlated, or empirical covariance estimation is unstable).

### Minor comments
5. Quotation around best on line 459 is not opened and closed properly.

**Questions:**

1. In LLM-as-judge, how representative is using a stronger LLM as “ground truth”?
2. What is the computational overhead for updating covariances at large $K$?
3. How does empirical covariance estimation error affect regret, particularly for small $N$ or weak correlation?

---

> ### Author Response · Authors · 2025-11-24
> **General Comment**
>
> We want to thank you for your valuable feedback. We would like to clarify and reiterate some of our contributions
> 1. We pose a new problem with this form of feedback and highlight its potential uses in online learning. This is a general conceptual contribution that can clearly be built upon in contextual bandits, RL, etc.
> 2. We perform a first theoretical treatment of this feedback structure, see how we can leverage the coupling to achieve better performance, and give some theoretical backing to the framework.
> 3. We provide numerical experiments showing the promise of this in synthetic experiments---albeit ones that use real data.
>
> We would also like to address technical contributions in our analysis. Since the covariance structure is unknown, we construct an empirical estimator $\lambda_t$ which is used in the AIPW estimator. Although the AIPW is a well-studied estimator, $\lambda_t$ being a random variable means that the concentration of the resulting AIPW estimator does not straightforwardly follow from Bennett’s concentration. It requires a careful covering argument along with a truncation built into the definition of $\lambda_t$, which ensures boundedness.

---

> ### Author Response · Authors · 2025-11-24
> **Justification of Assumptions 4.1 and 4.2 and discussion on robustness to misspecification of $\gamma$ parameter**
>
> Boundedness (Assumption 4.1) is a fairly common assumption in this space. The global variance ratio bound (Assumption 4.2) is an assumption that enables us to bound the random variable $\lambda_t$, thereby facilitating our analysis. This assumption also naturally represents the problem statement: coarse data is typically noisier, and it is therefore a reasonable assumption.
>
> We can, in fact, derive regret bounds for CUUCB without assumption 4.2, which allows us to see the loss in performance if the variance ratio $\gamma$ is mis-specified. Essentially, the assumption we really use in our proof is that $\frac{|\text{Cov}(i)|}{\text{Var}^\text{C}(i)} \le \gamma$. If we do not make this assumption, we only need to modify one step in the proof and obtain slightly modified regret bounds as described below.
>
> Define $\kappa(i):= \frac{|\text{Cov}(i)|}{\text{Var}^\text{C}(i)}$, and let
> $$\text{V}^\gamma(i):=
> \begin{cases}
> \text{Var}^\text{G}(i) - (1-\alpha)\gamma (2\kappa(i)- \gamma)\text{Var}^\text{C}(i), \text{ if }\gamma < \kappa(i) \\\\
> \text{V}^\*(i), \text{ otherwise}
> \end{cases}$$
> Then the regret bounds for CUUCB have the form:
>
> Instance-Dependent Regret Bound:
>      \begin{equation*}
>         \mathbb{E}[\text{Regret}(T)]
>         \le 128\sum_{i:\Delta_i>0} \frac{1}{\alpha(i)} \left(\frac{\text{V}^{\gamma}(i)}{\Delta_i} + b(1+\gamma +\kappa(i)) \right)\log(2T)
>      + \mathcal{O}(1),
>     \end{equation*}
>  Instance-Independent Regret Bound:
>      $$ \mathbb{E}[\text{Regret}(T)]
>         \le  23\sqrt{\left(\sum_{i} \frac{\text{V}^\gamma(i)}{\alpha(i)}\right)  T \log(2T)} + \mathcal{O}(\log(T))$$
> We can include a formal proof of the above results in a revised version as needed. We provide some intuition on the quantity $\text{V}^\gamma$ in the case when $\gamma < \kappa(i)$ for some arm $i$. For the remainder of the discussion, we drop the arm index $i$ for brevity.
>
> Intuitively, the above result is a consequence of the estimator $\lambda_t$ being truncated in the interval $[-\gamma', \gamma']$ (where $\gamma':=\gamma (1-\alpha_t)$). When $\gamma<\kappa$, this interval does not contain the optimal value $\lambda_t^\*$. Thus, the resulting minimum variance factor that is obtained in the concentration radius is given by
>
> $$\min_{\lambda \in [-\gamma', \gamma']} \text{V}_t^\lambda = \text{V}_t^{\gamma' \text{sgn(Cov)}} = \text{Var}^\text{G} - (1-\alpha_t)\gamma (2\kappa- \gamma)\text{Var}^\text{C},$$
>
> which has a limiting value $\text{V}^\gamma$ as the number of samples grows large. Thus, $\text{V}^\gamma$ replaces the optimal variance $\text{V}^\*$ in the regret bounds.
>
> Also note that as $\gamma$ approaches $\kappa$, we see that $\text{V}^\gamma$ approaches $\text{V}^\*$. Clearly, this bound demonstrates robustness to misspecification of $\gamma$, since we have:
> $$\text{V}^\gamma - \text{V}^\* = (1-\alpha)(\kappa-\gamma)^2\text{Var}^\text{C}.$$
> So if the misspecification of $\gamma$ with respect to the true value $\kappa$ is small, there is only a small increase in the variance. Also note that $\text{V}^\gamma \le \text{Var}^\text{G}$, which means we still obtain a variance reduction over using no coarse samples.
>
> In short, the misspecification of $\gamma$ causes CUUCB to use a clipped estimate of $\lambda^*_t$, which affects the resulting variance reduction in the manner described above.

---

> ### Author Response · Authors · 2025-11-25
> **Using "advanced LLM" as proxy for human labels**
>
> (In response to weakness 3) It is not necessary for the “advanced” LLM responses to precisely match human responses to carry out our experiment – the main idea is to show that CUUCB can debias the coarse feedback using a small amount of ground feedback, which is chosen to be the “advanced” LLM response here. The experiments serve as a proof of concept, since we do not have access to real human responses to the prompts. The ‘more advanced LLM’ is to be thought of as a model for human responses. The misalignment between the ‘advanced’ and ‘simple’ LLM response models the misalignment one might find between human and LLM responses. For some motivation on using LLMs to model human feedback, see the paper “Correlating and Predicting Human Evaluations of Language Models from Natural Language Processing Benchmarks” (Schaeffer et al., 2025).

---

> ### Author Response · Authors · 2025-11-25
> **Answering the Questions**
>
> 1. In LLM-as-judge, how representative is using a stronger LLM as “ground truth”?
>
> Response: Once again, this is just a modelling decision in the absence of true human feedback. The key objective is to show that CUUCB can correct misalignment between coarse and ground feedback. As added justification for this modelling decision, you can look at the paper “Correlating and Predicting Human Evaluations of Language Models from Natural Language Processing Benchmarks” (Schaeffer et al., 2025).
>
> 2. What is the computational overhead for updating covariances at large K?
>
> Response: This doesn’t scale with K. At a given time step, we only update covariances for a given arm.
>
> 3. How does empirical covariance estimation error affect regret, particularly for small N  or weak correlation?
>
> Response: That is a key part of the algorithm analysis and is what makes it non-trivial. Since we are working with covariance estimators, $\lambda_t$ is a random variable, and we require a careful covering argument to handle this fact in the proof of Lemma 5.2.

---

> > ### Comment · Reviewer_5jx2 · 2025-11-27
> >
> > I thank the authors for their detailed response. They have adequately addressed my comments and concerns and thus I am increasing my score.

---

### Author Response · Authors · 2025-12-04
**Message to AC : Remarks on discussion**

In the rebuttal, we generally emphasise the novelty of the framework and technical difficulties encountered due to the empirical estimator $\lambda_t$ when establishing regret guarantees. We justify our experimental setup using LLM-generated data in the absence of true human feedback. We address concerns regarding the robustness of CUUCB to misspecification of the $\gamma$ parameter in detail, which multiple reviewers found to be a useful addition. One of the reviewers was satisfied with our rebuttal and increased their score.

---

### Meta-Review · Area_Chair_hgrh · 2026-01-06

**Summary:**

This work studies a bandit problem with coupled uncertainties. In this setting, two types of sampling are considered: ground-truth data and coarse data linked by a coupling ratio. The main contribution is the derivation of a valid regret bound that depends on the number of ground-truth queries and the coupling ratio.

The reviewers are divided in their opinions. Positively, they acknowledge that the problem is well-motivated and that the regret bound represents a meaningful contribution. On the negative side, they note that the theoretical scope is confined to multi-armed bandits, the techniques lack novelty, and the experimental validation is insufficient.

Overall, I believe the limitations outweigh the merits of this work; therefore, I recommend rejection.

**Reviewer Concerns:**

Concerns addressed by the response:
1) The motivation regarding the usage of LLM is well explained;
2) Assumptions on the bound of variance are well clarified.


Concerns remaining after the response:
1) Limited theoretical scope ( this work only studies the multi-armed bandit problem);
2) Insufficient numerical experiments.

**Reviewer Scores:**

Reviewer 5jx2 agreed to increase his score. I expect other reviewers would have kept their score unchanged.

---

### Decision · Program_Chairs · 2026-01-26

Reject